# Optical and acoustic plasmons in the layered material Sr$_2$RuO$_4$

J. Schultz [1]✉, A. Lubk [1,2], F. Jerzembeck[3], N. Kikugawa [4], M. Knupfer[1], D. Wolf [1], B. Büchner [1,2] & J. Fink [1,2]✉

The perfect linear temperature dependence of the electrical resistivity in a variety of "strange" metals is a real puzzle in condensed matter physics. For these materials also other non-Fermi liquid properties are predicted or detected. In particular we mention the results derived from holographic theories which conclude that plasmons should be overdamped due to a low energy continuum in the electronic susceptibility. These predictions were supported by electron energy-loss spectroscopy in reflection on cuprates and ruthenates. Here we use electron energy-loss spectroscopy in transmission to study collective charge excitations in the layer metal Sr$_2$RuO$_4$. This metal has a transition from a perfect Fermi liquid below $T \approx 30$ K into a "strange" metal phase above $T \approx 800$ K. In this compound we cover a complete range between in-phase and out-of-phase oscillations. Outside the classical range of electron-hole excitations, leading to a Landau damping, we observe well-defined plasmons. The optical (acoustic) plasmon due to an in-phase (out-of-phase) charge oscillation of neighbouring layers exhibits a quadratic (linear) positive dispersion. Using a model for the Coulomb interaction of the charges in a layered system, it is possible to describe the range of optical plasmon excitations at high energies in a mean-field random phase approximation without taking correlation effects into account. In contrast, resonant inelastic X-ray scattering data show at low energies an enhancement of the acoustic plasmon velocity due to correlation effects. This difference can be explained by an energy dependent effective mass which changes from $\approx 3.5$ at low energy to 1 at high energy near the optical plasmon energy. There are no signs of over-damped plasmons predicted by holographic theories.

"Strange" metals are at present one of the most interesting research fields in solid state physics[1]. Due to the strong on-site interaction between their charge carriers, they show a deviation from a Fermi-liquid behavior, e.g., they do not show a quadratic but a linear temperature dependence of the electrical resistivity or linear in energy scattering rate in Angle-Resolved Photoemission Spectroscopy (ARPES)[2,3]. Moreover, there is no saturation at the Mott-Ioffe-Regel limit[4]. The unconventional and in some cases high-temperature superconductivity detected in these materials is supposed to be related to their "strange" normal-state electronic structure. Doped cuprates are prototypes of these "strange" metals. The non-Fermi liquid properties could be explained by a continuum of excitations up to an ultraviolet cutoff frequency $\omega_c$ in the electronic susceptibility[5], leading to a phenomenological marginal Fermi liquid theory. Integrating over

[1]Leibniz Institute for Solid State and Materials Research Dresden, Helmholtzstraße 20, 01069 Dresden, Germany. [2]TU Dresden, Institute of Solid State and Materials Physics, Haeckelstraße 3, 01069 Dresden, Germany. [3]Max Planck Institute for Chemical Physics of Solids, Nöthnitzer Straße 40, D-01187 Dresden, Germany. [4]National Institute for Materials Science, Tsukuba 305-0003, Japan. ✉e-mail: j.schultz@ifw-dresden.de; j.fink@ifw-dresden.de

this continuum yields a linear in energy imaginary part of the self-energy or scattering rate[6]. Electron Energy-Loss Spectroscopy (EELS) is a suitable experimental method to verify the existence of such a continuum because it measures the imaginary part of the electronic susceptibility $\Im\{\chi(\mathbf{q}, \omega)\}$. Here $\mathbf{q}$ is the momentum and $\omega$ is the energy. In nearly-free electron metals, there exists a $2k_F$ (with $k_F$ equal to the Fermi wave vector) wide stripe of a continuum of particle-hole excitations[7] which starts at $q \approx \omega/v_F$, where $v_F$ is the Fermi velocity.

In simple metals collective excitations (plasmons) exist below the critical wave vector $q_{crit} \approx \omega_P/v_F$ which is determined by the plasmon energy $\omega_P$ and $v_F$. Above this momentum the plasmons merge into the continuum and therefore are (Landau) damped by a decay into particle-hole excitations. Usually, $q_{crit}$ is between half and one Å$^{-1}$.

Surprisingly, early transmission EELS (T-EELS) studies of the highly correlated doped cuprates, using dedicated T-EELS spectrometers[8–10] with high momentum resolution, showed this behavior with a weak damping of plasmons below $q_{crit}$[11–16]. On the other hand, T-EELS studies using transmission electron microscopes with weak momentum resolution[17,18] showed no plasmon but a continuum. The difference can be easily explained by the poor momentum resolution in the TEM experiments ($\Delta q = 10$ and 30 Å$^{-1}$) which averages over the whole Brillouin zone (BZ) and therefore measures predominantly the Landau continuum above $q_{crit}$.

At variance with the early T-EELS measurements of collective excitations in hole doped cuprates great attention attracted the prediction of over-damped plasmons in "strange" metals and a replacement of these excitations by a continuum[19]. The authors point out that a novel theoretical model of strongly interacting matter may be necessary. They propose, that such a model would be potentially also related to high-$T_c$ superconductivity. The over-damping of plasmons is explained by holographic theories. Different from classical Landau damping at higher momentum the latter predict strongly enhanced damping also for long wavelength plasmons caused by quantum critical fluctuations. Recently, this work was supported by similar calculations[20].

Furthermore, there are several recent EELS experiments in reflection (R-EELS), supporting the theories which predict over-damped plasmons in "strange" or highly correlated metals. Only at very small momenta a well-defined plasmon exists followed by a transition into a featureless momentum-independent constant-in-frequency continuum well below $q_{crit}$[21–24]. Moreover, there is a very recent ARPES study on doped cuprates in which these holographic theories are supported by an asymmetric line shape at higher energies[25].

There are other differences between T- and R-EELS results from hole-doped cuprates: at small $q$, i.e., long wave length, T-EELS data show a positive dispersion which can be explained in RPA using an unrenormalized band structure[11,12,14,26]. On the other hand, R-EELS data show a negative dispersion, which may indicate a more localized electron liquid. The hybridization of the d-bands with the s-band in the alkali metals or many-body effects, when moving from Na to Cs is supposed to turn the plasmon dispersion from positive to negative[27,28]. On the other hand, the different result between T-EELS and R-EELS possibly can be explained by different response functions with respect to surface and bulk properties[29,30].

In this context, we mention that in various cuprates, well pronounced dispersive acoustic plasmons were detected by resonant inelastic X-ray scattering (RIXS)[31–33]. In all these measurements, weakly damped plasmons were detected for momentum ranging up to half of the size of the BZ.

For understanding the differences between T- and R-EELS on cuprates and to understand the influence of correlation effects on collective charge oscillation in general, we present here T-EELS data on

the related metal Sr$_2$RuO$_4$[34,35]. It is in some way intermediate between a normal Fermi liquid metal and a "strange" metal. Below $\approx 30$ K it is a perfect Fermi liquid which transforms at low temperatures $T_c = 1.5$ K into an unconventional superconductor[36]. It has other similarities to the cuprates: it has a perovskite structure formed by transition metal oxides layers. The essentially 2D correlated electronic structure is formed by three bands and has a van Hove singularity close to the Fermi level.

Deviating from the cuprate high-$T_c$ superconductors, it is a stoichiometric compound without crystallographic disorder due to dopant ions. Furthermore, the temperature dependence of the transport properties are more complicated. Above $T \approx 30$ K there is a crossover region in which Sr$_2$RuO$_4$ exceeds at $T_{MIR} \approx 800$ K the Mott-Ioffe-Regel limit, i.e., it turns to a "bad metal". These transport properties are partially related to Hund's rule coupling which causes strong correlation effects far from the insulating state[37].

Recently, we have studied the electronic structure of Sr$_2$RuO$_4$ by an investigation of the optical plasmon excitations with momenta parallel to the layers[38] using a dedicated T-EELS spectrometer[10]. Also in this highly correlated material, a well-defined plasmon could be detected near 1.5 eV. The plasmon has a positive dispersion and decays into a continuum of particle-hole excitations due to Landau damping, which could be explained in the framework of the random phase approximation (RPA) using an unrenormalized band structure.

Most of the previous momentum dependent EELS studies on layered materials were performed for a wave vector parallel to the layers. The reason for this is that thin samples (T-EELS) or clean surfaces (R-EELS) are easily prepared by a cleavage of the crystals parallel to the layers. In the present work, by focused ion beam milling, we are able to prepare a thin electron transparent lamella in which the layers are perpendicular to the surface. Using such samples, we could map out a complete set of plasmon excitations with momentum between parallel and perpendicular to the layers almost in the entire BZ. In this way, it is possible to control theoretical work on plasmon excitations in layered compounds, which is available since many decades[39–42].

At present, there is a strong discussion, whether spectroscopic results on the damping and dispersion of plasmons in "strange" metals can be explained on the basis of mean field theories such as RPA or whether we need new theories to explain valence band EELS results (see also the recent EELS review[43] which contrast the conflicting results of T-EELS and R-EELS). The present article strongly supports the results derived from T-EELS.

The dynamic structure factor is determined by the Fourier transformation of the charge density-density correlation function. It can be expressed[44] by the dynamical susceptibility $\chi(\mathbf{q}, \omega)$

$$S(\mathbf{q}, \omega) \propto \Im\{\chi(\mathbf{q}, \omega)\} \propto \Im\left\{-\frac{1}{\epsilon(\mathbf{q}, \omega)}\right\}. \tag{1}$$

Here, $\epsilon(\mathbf{q}, \omega)$ is the complex dielectric function.

The calculation of the Lindhard-Ehrenreich-Cohen susceptibility of the many-body system of the charge carriers in solids is a challenging task. The susceptibility $\chi_0$ for a non-interacting one-band electron liquid is given by[14,45]

$$\chi_0(\omega, \mathbf{q}) = \int_{BZ} M(\mathbf{q}, \mathbf{k}) \frac{2F(\mathbf{k})\Delta E(\mathbf{q}, \mathbf{k})}{(\omega + i\Gamma)^2 - \Delta E^2(\mathbf{q}, \mathbf{k})} d^3k. \tag{2}$$

Here $\Delta E = E_{\mathbf{k}+\mathbf{q}} - E_{\mathbf{k}}$, $E_{\mathbf{k}}$ are the band energies of the electrons having a momentum $\mathbf{k}$, $M(\mathbf{q}, \mathbf{k})$ is related to matrix elements, $\Gamma$ is the lifetime broadening of the particle-hole excitations, and $F(\mathbf{k})$ is the Fermi function.

While $\chi_0$ is the Lindhard-Ehrenreich-Cohen susceptibility for single-particle excitations related to an external field, stemming from the field of the scattering electron, $\chi$ is the susceptibility for the total field, including the induced one. Running a self-consistency cycle, we obtain in the mean field RPA the result

$$\chi^{\text{RPA}}(\mathbf{q}, \omega) = \frac{\chi_0(\mathbf{q}, \omega)}{\epsilon_b - V(\mathbf{q})\chi_0(\mathbf{q}, \omega)}. \tag{3}$$

Here, $V(\mathbf{q})$ is the Fourier transformed Coulomb interaction between the charge carriers and $\epsilon_b$ is the background dielectric constant. The dielectric function can be calculated from

$$\epsilon(\mathbf{q}, \omega) = \epsilon_b - V(\mathbf{q})\chi_0(\mathbf{q}, \omega). \tag{4}$$

In this approximation and for small damping, there are in addition to the single-particle excitations collective excitations, termed plasmons. The energy of the plasmon is determined by the zeros of the real part of the denominator of Eq. (3). The long wavelength energy of the plasmon in the RPA is given by

$$\omega_{\text{P}}(0)^2 = \frac{4\pi N e^2}{\epsilon_b m^*} \tag{5}$$

with $N$ being the density of the charge carriers, $m^*$ the effective mass and $e$ the elementary charge.

For small but finite momentum, up to $q^2$ and within the RPA, the dispersion is given by

$$\omega_{\text{P}}(q) = \omega_{\text{P}}(0) + A_{\text{RPA}}q^2 + \dots; \quad A_{\text{RPA}} = (A_1 + A_2). \tag{6}$$

$A_1$ is related by the finite compressibility or the squared averaged Fermi velocity of the electron liquid and is always positive. $A_2$, which is always negative, is proportional to the size of the effective mass[26].

For free-electron metals only

$$A_1 = \frac{1}{\omega_{\text{P}}(0)}\frac{3}{10}\langle v_{\text{F}}^2 \rangle_{\mathbf{q}} \tag{7}$$

determines the optical plasmon dispersion. Here, the averaged squared Fermi velocity along the $\mathbf{q}$ direction is defined by[14]

$$\langle v_{\text{F}}^2 \rangle_{\mathbf{q}} = \left\langle \left(\frac{\mathbf{q}}{\hbar q}\frac{\partial E_{\mathbf{k}}}{\partial \mathbf{k}}\right)^2 \right\rangle. \tag{8}$$

For metals where the band dispersion is strongly reduced by a finite effective mass enhancement $m^*/m_O$, the negative $A_2$ may dominate the optical plasmon dispersion, leading in total to a negative dispersion[26].

In the following we discuss the structure factor V($\mathbf{q}$). In a homogeneous electron system

$$V(\mathbf{q}) = \frac{4\pi e^2}{q^2}. \tag{9}$$

For a system, built up by 2D layers separated by the distance $d$ we use the Fetter model with the Coulomb potential

$$V(\mathbf{q}) = V(q_{\parallel}, q_{\perp}) = \frac{4\pi e^2}{q^2}\frac{q_{\parallel}d}{2}\frac{\sinh(q_{\parallel}d)}{\cosh(q_{\parallel}d) - \cos(q_{\perp}d)}, \tag{10}$$

where $q_{\parallel}$ ($q_{\perp}$) is the momentum parallel (perpendicular) to the layers[40]. Therefore, in a layered system, the plasmon dispersion is not only determined by the compressibility and by the effective mass of the electron liquid (see Eq. (6)) but also by the structure factor $V(\mathbf{q})$,

depending on $q_{\parallel}d$ and $q_{\perp}d$

$$\omega_{\text{P}}(q)^2 = s^2 q^2 + \omega_{\text{P}}(0)^2\frac{q_{\parallel}d}{2}\frac{\sinh(q_{\parallel}d)}{\cosh(q_{\parallel}d) - \cos(q_{\perp}d)} \tag{11}$$

with $s^2 = \frac{1}{2}\langle v_{\text{F}}^2 \rangle_{\mathbf{q}}$. In the low $q$ approximation Eq. (11) yields again a quadratic plasmon dispersion

$$\omega_{\text{P}}(q) = \omega_{\text{P}}(0)\left(1 + \frac{1}{2}\left[\frac{1}{12}d^2 + \frac{s^2}{\omega_0(0)^2}\right]q_{\parallel}^2\right). \tag{12}$$

For a dielectric function that is consistent with the above dispersion relation we refer to the Method section. For small $q_{\parallel}$ and $q_{\perp}d = 0$, the structure factor $V(q_{\parallel}, q_{\perp})$ is the same as in the homogeneous electron gas. In this case, the charge oscillations in the layers are in phase and the plasmon dispersion is the same as in a homogeneous 3D electron system. For $q_{\perp}d = \pi$ the charge oscillations between neighboring layers are out of phase.

This leads to an acoustic plasmon, with a rather small energy gap due to the layer interaction[46] in the long wavelength limit. The dispersion of the acoustic plasmon for small $q_{\parallel}$ is given by

$$\omega_{\text{P}}(q_{\parallel}) = q_{\parallel}\sqrt{s^2 + \omega_{\text{P}}(0)^2\frac{d^2}{4}}. \tag{13}$$

Very often, the first term in the square root is considerably smaller than the second. Then the phase velocity of the acoustic plasmon along the direction of $q_{\parallel}$ is given by

$$v_{\text{p}} = \frac{\omega_{\text{P}}(0)d}{2}. \tag{14}$$

Besides the hydrodynamic Fetter model, the plasmon dispersion of a layered system was also derived by means of RPA, leading to similar results[47].

## Results

T-EELS measurements were performed on single-crystalline $Sr_2RuO_4$ at room temperature. Figure 1 shows the crystal structure of $Sr_2RuO_4$ which consists of $RuO_2$ layers stacked with SrO spacer layers along the c-axis direction. The thin films were characterized by in-situ electron diffraction and the crystallographic axes were oriented with respect to the (a, c)-plane (see Fig. 2a). The distance $d = 6.36$ Å between the $RuO_2$ layers is half the c-axis lattice constant. The thin lamellas for T-EELS with a normal direction parallel to the b-axis were cut with a focused ion beam. In this way, EELS experiments were possible with the momentum parallel to the (a, c)-plane. We emphasize that this momentum range is different from our previous EELS study on $Sr_2RuO_4$[38] where we covered the momentum range in the (a, b)-plane.

T-EELS was performed with a primary electron energy of 80 keV and with an energy and momentum resolution of 120 meV and 0.04 Å$^{-1}$, respectively. The momentum-resolved EELS data were sequentially recorded in the ($q_{\parallel} = q_a$, $q_{\perp}$) momentum plane as depicted in Fig. 2b. In Fig. 3 we show typical EELS intensities as a function of the energy for various $q_{\parallel}$ and $q_{\perp}$ values. With increasing $q_{\parallel}$ the plasmon energy slightly increases, whereas for increasing $q_{\perp}$ the plasmon energies decrease. At low total momentum it is difficult to see a well defined plasmon due to the high intensity of the quasi-elastic peak. The same holds for high total momentum because the T-EELS cross section is decreasing with $1/q^2$ (see below). Except for the described cases, well developed and dispersing plasmon excitations below 1.8 eV are visible. From the fit to the loss data with a Drude function, we obtain the energy, the width, and the intensities (see Methods).

The energy is plotted in Fig. 4 as a function of $q_{\parallel}$ for various $q_{\perp}$ values together with theoretical curves calculated in the framework of

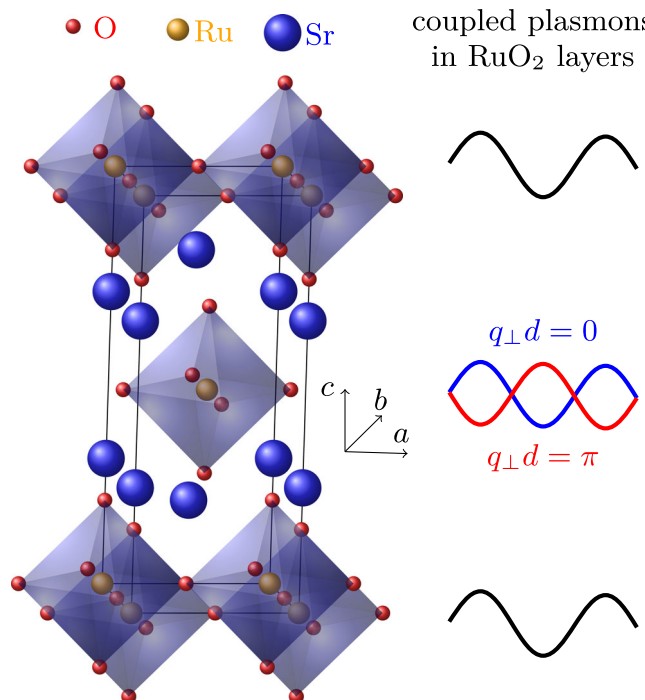

**Fig. 1 | Schematic of the charge oscillations in the layered material.** Crystal structure of $Sr_2RuO_4$ (left hand site) and schematic representation of in-phase and out-of-phase two dimensional charge oscillations (right hand site).

the Fetter model (see Section II). We use the parameters $\omega_p(0) = 1.48$ eV (from optical spectroscopy[48]), $d = 6.36$ Å, and $\langle v_F^2 \rangle_{100} = 4.91$ (eV Å)$^2$. The latter value for the three Ru 4d $t_{2g}$ bands crossing the Fermi level was derived from a tight-binding (TB) band structure[49] (see Methods). For all plasmon energies above the continuum, within error bars, there is rather good agreement between theory and experiment. There is a continuous transition between the optical plasmon for $q_\perp = 0$ (purple data) at higher energy to the acoustic plasmon ($q_\perp = 0.4$ Å$^{-1}$) close to $q_\perp = \pi/d = 0.49$ Å$^{-1}$ (red data) at lower energy. Due to our finite energy resolution we cannot follow the acoustic plasmon to zero energy. Furthermore, the momentum resolution of the instrument is limited by the finite width of the collection aperture of our spectrometer. Since the signal is integrated over the latter we observe a drop of the plasmon energy also for the optical plasmon near $q_\perp = 0$ (see purple data in Fig. 4).

The difference between extrapolated RPA plasmon dispersion and experimental data for $q_\parallel \gtrsim q_{crit}$ (see Fig. 4) can be explained by Landau damping. For these wave vectors, the plasmon decays into over-damped plasmon excitations and into spectral weight which is caused by the Lindhard continuum. This leads to a reduction of the energy of the maximum in the total loss function (see Fig. 5). This reduction has been observed in our calculations presented in ref. 38. A similar behavior has been also observed for Al[50]. In Fig. 4 we also show the continuum of the single-particle intra-band transitions $\chi_0$ (see the gray region) calculated using an unrenormalized TB band structure[49] (see Methods). The optical plasmon merges into the continuum near $q_\parallel \approx 0.4$ Å$^{-1}$. We assign this value to the critical momentum $q_{crit}$. This value is further corroborated by the rapid increase of the plasmon width at this wave vector (see Fig. 6).

## Discussion
### Optical and acoustic plasmons in layered "strange" metals
The data presented in Fig. 3 demonstrate that T-EELS in layered systems is capable of probing not only in-phase (optical) plasmons with momentum parallel to the layers but also out-of phase (acoustic)

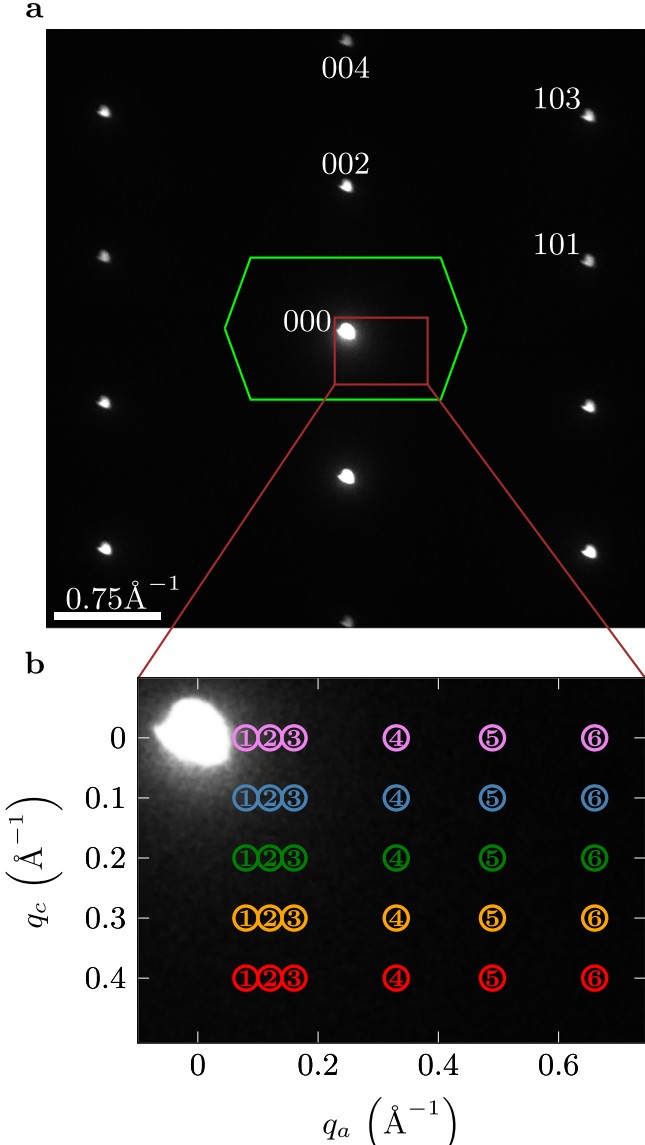

**Fig. 2 | Electron diffraction on the $Sr_2RuO_4$ crystal. a** Indexed electron diffraction pattern (white dots) in the ($q_\parallel$, $q_\perp$) plane, including the reduced Brillouin zone (green) for equal layers along the $c$-axis. **b** Brown momentum range [see (**a**)] in which loss spectra are recorded for various $q_\parallel$ and $q_\perp$ values. The colors purple, blue, green, yellow, and red correspond to $q_\perp$ equal 0, 0.1, 0.2 0.3, and 0.4 Å$^{-1}$, respectively. The diameter of the filter entrance aperture [indicated by the colored circles in (b)] defining the momentum range in one EEL spectrum (momentum resolution) corresponds to a momentum of 0.04 Å$^{-1}$.

collective charge oscillations in neighboring layers. A suitable sample preparation is important for such studies. Thus, we show that T-EELS can compete with recent RIXS studies of acoustic plasmons of cuprates[31,32,46]. This is an important extension of T-EEL spectroscopy. Furthermore, we emphasize that the present work shows that acoustic plasmons exist also in non-cuprate "strange" metal layer systems.

Well defined plasmons exist in the complete momentum range which is not covered by the range of single-particle excitations calculated in the mean-field RPA theory. Optical plasmons exist in $\approx 15\%$ of the BZ. The rest is determined by a continuum of intra-band single particle excitations, which strongly dampen the plasmon excitation in excellent agreement with previous studies[38]. There is no sign of a reduction of the coherent plasmon range due to an over-damping discussed in terms of holographic theories[19,20].

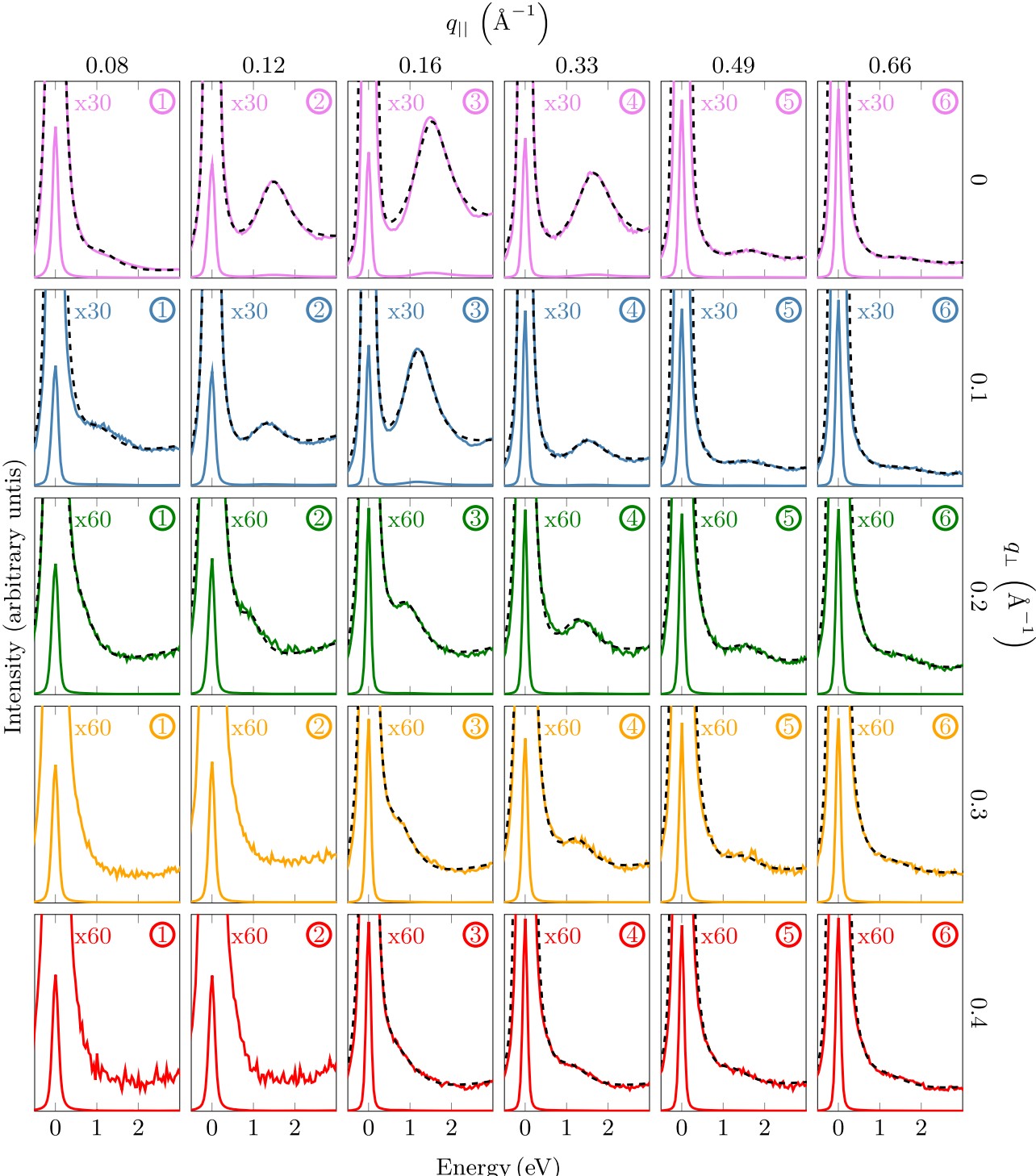

**Fig. 3 | Electron energy-loss intensities for various $q_{||}$ and $q_{\perp}$ values.** The indicated numbers (1–6) and the colors correspond to the momentum values depicted in Fig. 2. Experimental spectra are depicted twice (normal scale + 30x/60x scaled up) to show both the zero loss peak and the plasmon peak. The black dashed lines correspond to fits of a superposition of a Voigt profile (zero loss), a Drude function (plasmons), and a background (see Methods).

Next we discuss the dispersion of the optical plasmon in the momentum range $q_{||} \lesssim q_{crit} \approx 0.4$ Å$^{-1}$ (see Fig. 4 purple data). It can be well fitted by Eqs. (1) and (12), relations which are derived by a mean-field theory in the Fetter-Apostol model[40,47]. We obtain a dispersion coefficient $A_{exp} = 2.1 \pm 0.2$ eV Å$^2$. Calculating the dispersion coefficient by Eq. (12) using $\langle v_F^2 \rangle_q = 4.91$ (eV Å)$^2$ we obtain $A = 2.8$ eV Å$^2$. The small difference between experimental and calculated dispersion coefficients can be explained e.g. by a 1.5 Å thickness of the RuO$_2$

layers which reduces the nominal $d$ to $d_{red} = 4.85$ Å. This reduction brings the theoretical $A$ value very close to the experimental one. Moreover, our calculations show that an enlargement of the half width of $\Gamma$ from about 0.1 to the experimental value of $\approx 1$ eV reduces the dispersion coefficient $A$ by $\approx 0.2$ eV Å$^2$. Thus the experimental optical plasmon dispersion can be well described by an unre-normalized band structure. Moreover, using a Fermi velocity which is reduced by an effective mass $m^*/m_0 = 3.5$, we derive a corresponding

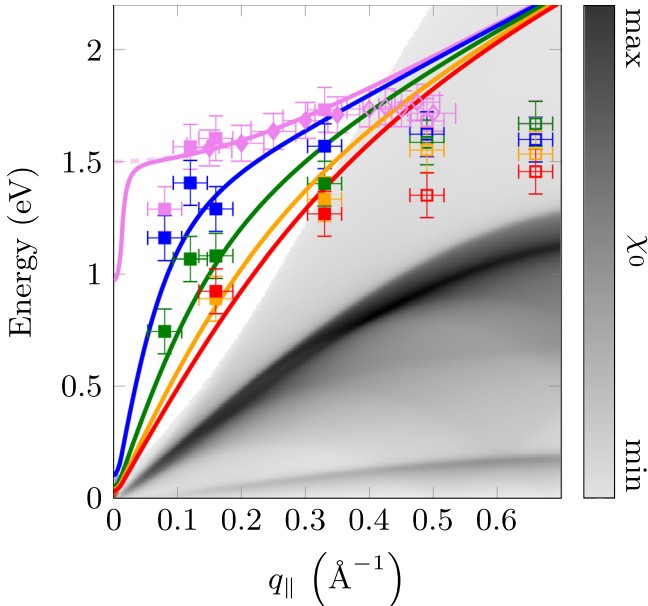

**Fig. 4 | Plasmon dispersion along the momentum $q_{\parallel}$ parallel to the layers for several $q_{\perp}$ perpendicular to the layers (squares) together with calculations within the framework of the Fetter model (solid lines).** The $q_{\perp}$ values 0, 0.1, 0.2, 0.3, and 0.4 Å$^{-1}$ are marked by purple, blue, green, yellow, and red color symbols, respectively (see Fig. 3). We have added also the data from the optical plasmon dispersion derived in our previous publication (dark purple diamonds)[38]. The horizontal error bars originate from the finite momentum resolution while the vertical ones are related to the finite spectral resolution of the EEL spectrometer (the fitting error is small in comparison). The region marked in gray corresponds to the susceptibility $\chi_0$ calculated from a tight binding band structure (see below and Methods). The excitations in the continuum range are marked by open symbols. The colored solid (dashed) curves correspond to theoretical data considering (neglecting) finite momentum resolution, i.e., integration within the EEL collection aperture.

---

dispersion coefficient $A = 0.8$ eV Å$^2$ which is at variance with the experimental data.

In Fig. 5 we compare the T-EELS data of the optical plasmon dispersion with those derived from theoretical calculations of the loss function [Eq. (1)]. For the susceptibility [Eq. (2)] we use the tight binding band structure from ref. 49. The dielectric function is derived using Eq. (4) together with the Coulomb potential of a layered electronic system given in Eq. (10). A small $\Gamma = 0.1$ eV is used to better visualize the small dispersion. In Fig. 5 we present the results for the susceptibility, the loss function, and the optical plasmon dispersion for an unrenormalized band structure ($m^*/m_0 = 1$), a constant effective mass ($m^*/m_0 = 3.5$), and an energy dependent effective mass ranging from 3.5 at low energies to 1 above ≈ 0.2 eV. The latter was taken from optical spectroscopy[48] together with an extrapolation to 1 at higher energies.

For the unrenormalized band structure ($m^*/m_0 = 1$) and for the one, which is renormalized at low energies only, the calculations predict a classic plasmon dispersion, which merges at $q_{crit} \approx 0.4$ Å$^{-1}$ into the Lindhard continuum (see columns 2 and 3). In both cases (see row 1 and 3, column 4) the calculated dispersion agrees well with the experimental one. However, for the renormalized band structure, using the low-energy average $m^*/m_0 = 3.5$ from optical spectroscopy[48], the calculated data deviates considerably from the experimental one. The continuum is strongly lowered in energy, preventing the merging of the plasmon into a continuum and hence as a strong increase of the peak width above $q_{crit}$. Moreover, the plasmon dispersion is reduced, in disagreement with the experiment. The effective mass dependence of the plasmon dispersion was already discussed in our previous paper[38].

Using the energy dependent effective mass from optical spectroscopy below $\omega = 0.2$ eV plus an extrapolation to $m^*(\omega)/m_0 = 1$ at higher energies, we see the expected strong renormalization and an intensity increase in the low energy/momentum range. However, the susceptibility is similar to that calculated from an unrenormalized band structure at higher energy/momentum. This yields a plasmon dispersion, which is very close to an unrenormalized dispersion. Thus our present and the previous experimental data of the dispersion of the optical plasmon indicates that the long-wavelength dispersion can be explained in the framework of a mean-field RPA theory using an effective mass of one. Because an energy dependent effective mass is expected also for the cuprates, the present result can potentially also explain the unrenormalized plasmon dispersion detected in the cuprates[14,26].

The calculations clearly demonstrate that the low energy renormalization of the susceptibility/optically conductivity does not transfer into the high-energy plasmon dispersion. A similar behavior was predicted for the case of electron-phonon coupling[51]. This was also discussed in a standard solid state text book[52] where it was stated that well above the Debye energy phonons do not renormalize the band structure. On the other hand, the renormalized acoustic plasmon dispersion observed by RIXS at low energies in cuprates[53] can be explained in this framework of a renormalized band structure.

In Fig. 6 we present the optical plasmon width as a function of $q_{\parallel}$. We show data derived from Fig. 3 and from our previous EELS experiments[38]. Within error bars there is a good agreement between the two datasets. The plasmon width at zero momentum is smaller than the plasmon energy, indicating a coherent collective charge excitation. The width below $q_{\parallel} \approx 0.4$ Å$^{-1}$ is nearly constant. Near $q_{\parallel} = 0.4$ Å$^{-1}$ the width increases, indicating the merging of the plasmon dispersion into the single particle continuum at a $q_{crit} \approx 0.4$ Å$^{-1}$ (See also Fig. 4). We compare the experimental results with those derived from the calculated loss function. For the effective mass equal to 3.5 there is no merging of the plasmon line into the continuum because the latter is strongly lowered in energy (see also Fig. 5). On the other hand, the calculation for an unrenormalized effective mass using an offset of 1 eV is in qualitative agreement with the experiment. The offset will be explained in detail below. The difference in $q_{crit}$ of 0.1 Å$^{-1}$ may be caused by the absence of spin-orbit interaction and matrix elements in the calculations.

This indicates again that at the relative high plasmon energy, the effective mass is close to one and supports the formation of resilient quasi-particles[48] which were predicted by DFT+DMFT calculations[54].

Already in our previous paper[38], we pointed out that the plasmon width below $q_{crit}$ is nearly constant. From this we conclude that the plasmon width in our system is not related to electron-electron interaction, since for the latter a quadratic increase of the width, starting at zero energy, would be expected with increasing momentum transfer[55]. Furthermore, the finite width at zero energy cannot be explained in terms of a temperature dependent scattering rate, proportional to the imaginary part of the self-energy ($\Im\Sigma$). Recent ARPES experiments on $Sr_2RuO_4$[56] report near 20 K $\Im\Sigma \approx 0.01$ eV and at room temperature $\Im\Sigma \approx 0.15$ eV. These values are well below the observed width of 1 eV. Essentially, no temperature dependence of the width has been detected in our previous experiments[38]. In addition, the finite width at zero energy is likely not caused by electron-phonon interaction[57] nor by impurity scattering (different for superconducting cuprates, $Sr_2RuO_4$ is a stoichiometric compound without doping ions).

A smaller width in comparison to the energy hints that the overdamping is not caused by quantum critical fluctuations which is in conflict with theoretical predictions[19]. In fact, it is caused by a decay into interband transitions[27,58–60], as it is the case for the majority of metallic systems investigated by T-EELS. The origin of these interbands are back-folded bands from the second to the first BZ due to a finite

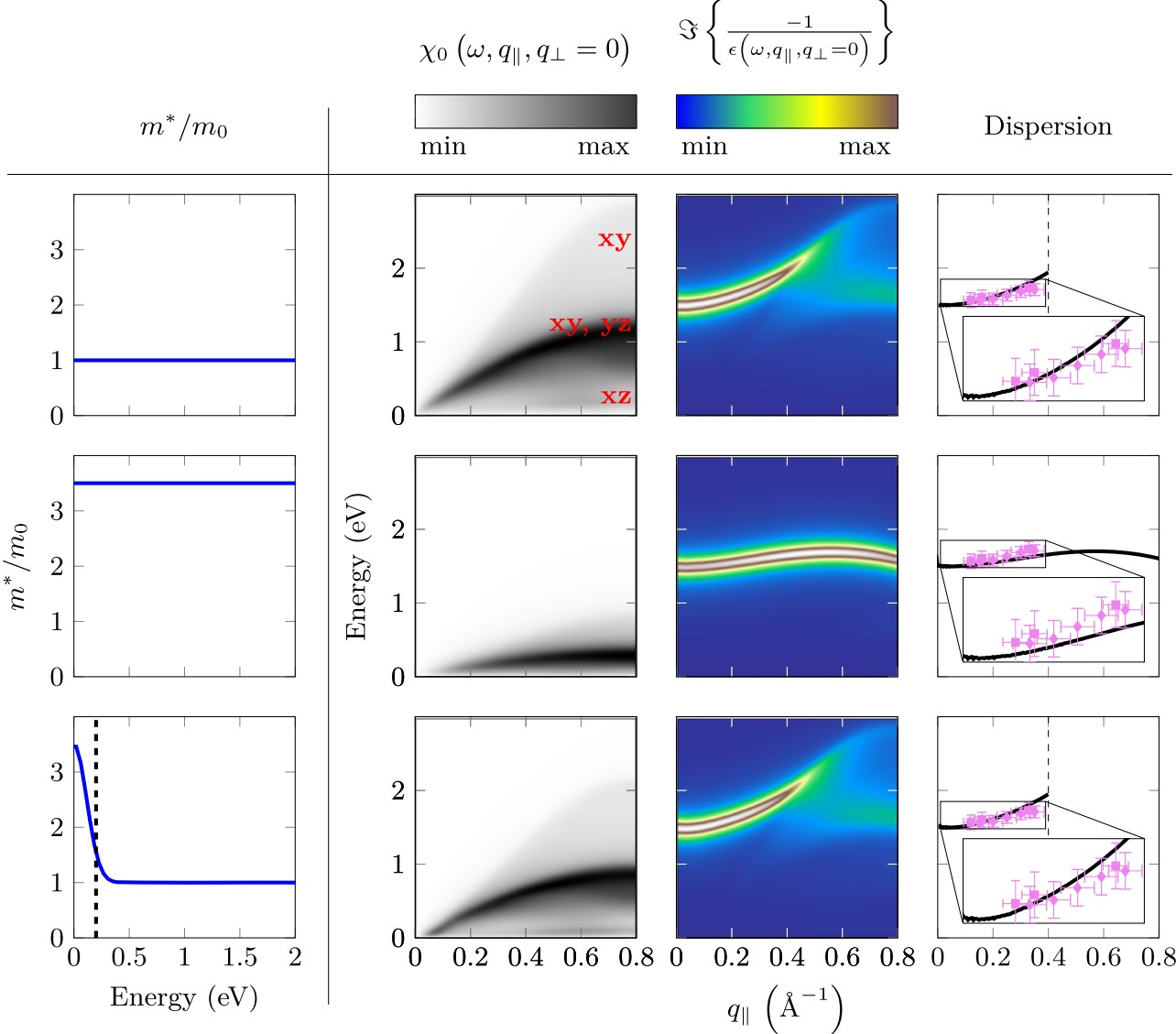

**Fig. 5 | Calculated imaginary part of the susceptibility $\chi_0(\omega, q_\parallel, q_\perp = 0)$ (second column), loss function $\Im\left\{-\frac{1}{\epsilon(\omega, q_\parallel, q_\perp = 0)}\right\}$ (third column), and plasmon dispersion (fourth column) along the momentum ($q_\parallel, q_\perp = 0$).** In the calculation various effective masses were used: first row $m^*/m_0 = 1$, second row $m^*/m_0 = 3.5$, and an energy dependent effective mass $m^*(\omega)/m_0$ shown in the third row. The red labels in the susceptibility in the first row are related to the origin of the three electronic bands in $Sr_2RuO_4$. In column four we compare the experimental optical plasmon dispersion (shown in Fig. 4) with the calculated one for $m^*/m_0 = 1$ and 3.5, and a band structure which is renormalized only at low energies. The horizontal error bars originate from the finite momentum resolution while the vertical ones are related to the finite spectral resolution of the EEL spectrometer.

pseudo-potential. In recent RIXS data on p- and n-typed cuprates[53], the difference in acoustic plasmon width could be described in this framework.

In Fig. 7 we depict the plasmon intensities as a function of $q_\parallel$ and $q_\perp$ (see Methods section for details of the evaluation). The decay of the total spectral weight of the plasmon resonances at large momentum transfers is approximately proportional to $q^{-2}$ as observed for conventional bulk plasmons and predicted by the longitudinal f-sum rule and the theoretical dielectric function of the Fetter model (see Methods). The decay observed for $q_\perp \neq 0$ in the long wavelength limit is also consistent with the theoretical dielectric response as well as a version of the long wavelength sum rule implying that the full spectral weight at $q = 0$ is concentrated in the bulk plasmon $q_\perp = 0$ mode (see Methods and ref. 61 for the sum rules). The shift of the maximal integrated loss intensity towards smaller momentum transfer predicted by theory may be attributed to shortcomings of the Fetter model dielectric

function (see Methods), impact of the zero loss, and limited experimental resolution.

In the following we discuss the acoustic plasmon data (see red data and line in Fig. 4). Despite the gap at low energy due to a finite energy resolution, the dispersion extrapolates to zero energy typical of an acoustic plasmon. The derived experimental plasmon velocity is $v_P \approx 4.7$ eV Å. From Eq. (13) we derive $v_P \approx 4.8$ eV Å in very good agreement with the experimental value. This indicates that the first term in Eq. (13) due to the finite Fermi velocity is small compared to the term which only depends on $\omega_P(0)$ and $d$. Thus, for a given $\omega_P(0)$ the acoustic plasmon dispersion only depends on $d$ and is not influenced by a large Fermi velocity but hints to a reduced one due to correlation effects (enhanced effective mass). Unfortunately, the finite energy resolution in the present T-EELS experiment does not allow a quantitative determination of the effective mass at low energies. However, low-energy RIXS data on cuprates indicate, that an enhanced effective

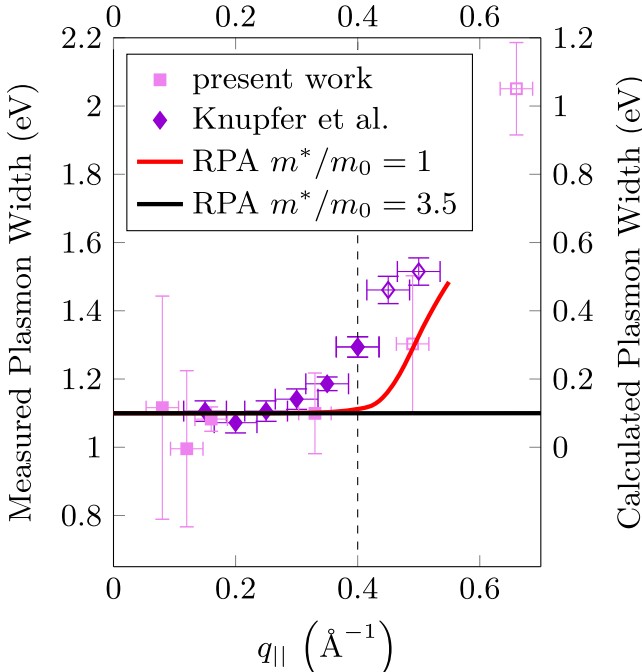

**Fig. 6 | Plasmon width at half maximum in dependence on the momentum transfer parallel to the layers at $q_\perp = 0$, determined from a fit to the EELS data (see Methods).** The horizontal error bars originate from the finite momentum resolution while the vertical ones are related to the residual error of the fit. For comparison the corresponding data from a Drude fit of our previous EELS measurements reported in ref. [38] was added as dark purple diamonds. The energy of the maximum of the excitations in the continuum range (above $q_\parallel \approx 0.4$ Å⁻¹) are marked by open symbols. The experimental values are compared with data derived from RPA calculations of the loss function for an unrenormalized (red line) and a renormalized (black line) band structure. For the direct comparison, the y-axis corresponding to the calculated width is shifted by 1 eV to recognize decay into interband transitions, not included in the calculations of the loss function. Furthermore, the width of the unrenormalized curve is evaluated for $q_\parallel < 0.55$ Å⁻¹ only due to splitting of the spectral weight into a damped plasmon and a Lindhard continuum above $q_{crit}$ (see Fig. 4 upper row, third column).

mass is necessary to describe the plasmon velocity of the acoustic plasmons[53].

Neglecting the Fermi velocity term, the linear acoustic plasmon dispersion [see Eq. (13)] can be explained in the following way. The phase difference π of the oscillations between neighboring layers reduces the plasmon energy from $\omega_P(0)$ to zero. When adding a momentum $q_\parallel$ the phase difference between neighboring layers is increased to $\pi + q_\parallel d$ and therefore, using a linear relation, the energy of the acoustic plasmon should increase by $(\omega_P(0)dq_\parallel)/\pi$ which is close to Eq. (14).

In summary, the long wavelength $q_\parallel$ dispersion of the $q_\perp$ dependent plasmons including the optical and the acoustic collective excitations and the decay of the optical plasmon by Landau damping can be all explained in terms of a mean-field RPA model. It is possible to understand this interpretation of the present results in the following way. Long wavelength charge excitations are not influenced by local interactions such as on-site Coulomb and Hund's exchange interactions. This behavior is different from ARPES studies, in which local properties play an important role. In this context it is also important to note that monopole (a single hole) excitations detected in ARPES are differently screened compared to dipole excitations recorded in EELS. We further emphasize that our present analysis of the acoustic plasmon dispersion is also important for the understanding of previous[31,32,46] and future RIXS studies on cuprates.

## Perspectives

The present study has demonstrated that optical and acoustic plasmons can be investigated by T-EELS in the complete BZ in layered systems. Therefore, with the advent of higher energy resolution, T-EELS will be competitive at lower energies with RIXS, also taking into account that momentum-resolved T-EELS provides a direct probe of the dynamic susceptibility. It will be possible to study in more detail the different influence of correlation effects on optical and acoustic plasmons, caused by an energy dependent effective mass. The latter was predicted by a combined density functional/dynamical mean-field theory (DFT + DMFT) calculation[54] and experimentally detected by optical spectroscopy[48]. Furthermore, in "strange" metals, it will be possible to study low-energy and high-momentum charge excitations which were predicted in ref. [62] to depend on correlation effects. In this way it will be possible to evaluate the spatial dependence of the density-density fluctuations in "strange" metals.

## Methods

### Dielectric response of the layered plasmon system

The dielectric function corresponding to the Fetter model of a system of coupled 2D layers reads[40]

$$\epsilon(\omega, q_\parallel, q_\perp) = 1 - \frac{2\pi N e^2 q_\parallel / m}{\omega(\omega + i\Gamma) - s^2 q_\parallel^2} \frac{\sinh(q_\parallel d)}{\cosh(q_\parallel d) - \cos(q_\perp d)}. \quad (15)$$

This dielectric function is an approximation assuming a perturbation charge that is confined to the 2D layers supporting the plasmons, which is violated by the electron beam resulting in deviations to the experimental dielectric response. However, the intensities of the plasmon peaks can be derived from this dielectric function by calculating the loss probability (T-EELS signal) using Eq. (1) and integration along $\omega$ (see Fig. 7). A version of the long wave wavelength sum rule for the dynamic susceptibility reads[61]

$$\lim_{q \to 0} \Im\left\{ \frac{1}{\epsilon(q, \omega)} \right\} = -\frac{\pi \omega_P}{2} \left( \delta(\omega - \omega_P) - \delta(\omega + \omega_P) \right). \quad (16)$$

Thus, in the $q \to 0$ limit, the dynamic susceptibility is determined by the longitudinal plasmon.

### Samples

Sr$_2$RuO$_4$ crystals were grown using the floating-zone method[63]. The superconducting transition temperature of the sample was $T_c = 1.5$ K.

Thin TEM lamellas of Sr$_2$RuO$_4$ with the normal pointing along the b-axis were prepared by Focused Ion Beam (FIB) using a Thermofisher instrument. The target thickness of the lamellas was 80 nanometers. Low ion energy polishing was used as final step to thin Ga-ion damaged surface layers.

### EELS measurements

The momentum-resolved loss function was recorded at a non-monochromized Hitachi HF3300S at 60 kV acceleration voltage (pre-characterization) and a FEI Titan³ TEM equipped with a Wien monochromator and a Gatan Tridiem imaging filter (GIF) at 80 kV acceleration voltage in a serial way (i.e., one EEL spectrum per fixed momentum). The energy resolution is around 120 meV (FWHM of zero loss peak). At a camera length (i.e., effective distance between sample and detector) of 1.15 m, the GIF entrance aperture was used to select the different momentum values and covers a momentum range of 0.04 Å⁻¹ (0.13 mrad semi collection angle). The acquisition times are presented in Table 1.

Due to instabilities of the monochromator and the rather long collection times required at large momentum transfers, the recorded spectra are subject to substantial noise as well as mutual random fluctuations/offsets. In order to mitigate these effects, each spectrum

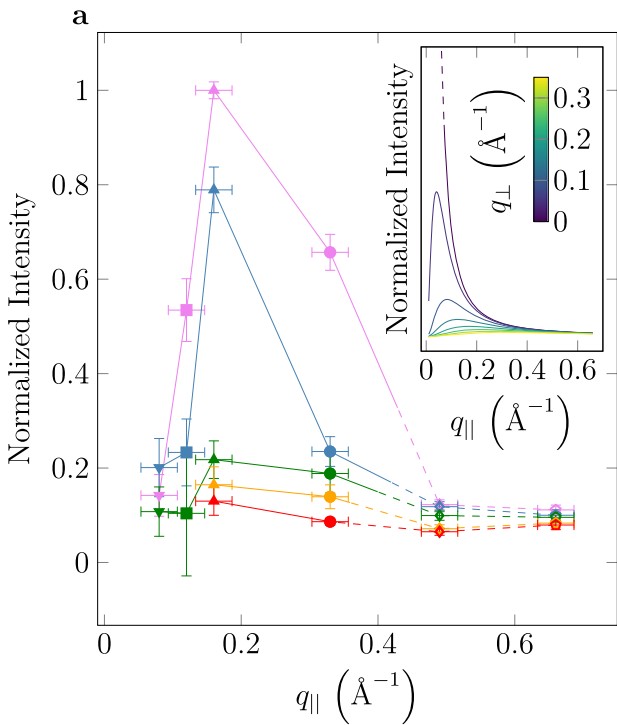

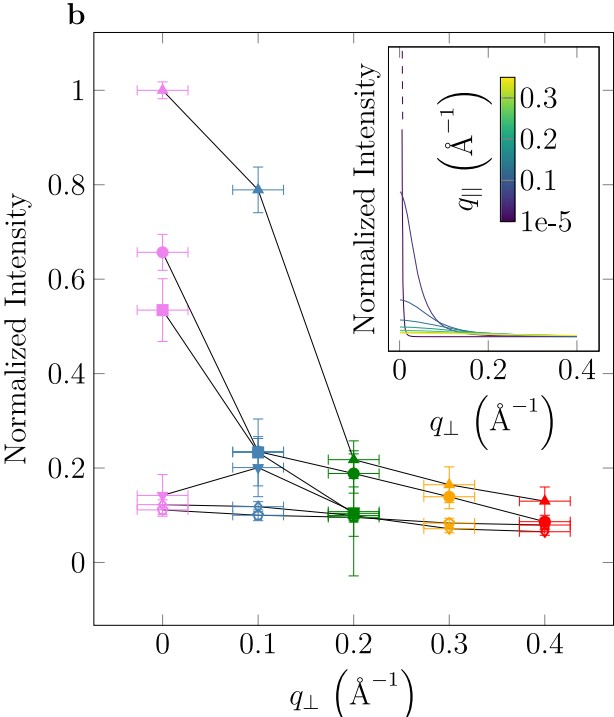

**Fig. 7 | Integrated loss intensities under the fitted plasmon peaks as a function of in-plane ($q_{||}$) and out-of-plane ($q_\perp$) momentum transfer.** The horizontal error bars originate from the finite momentum resolution while the vertical ones are related to the residual error of the fit. The $q_\perp$ values 0, 0.1, 0.2, 0.3, and 0.4 Å$^{-1}$ are marked by purple, blue, green, yellow, and red color symbols, respectively (see Fig. 3). The $q_{||}$ values 0.08, 0.12, 0.16, 0.33, 0.49, and 0.66 Å$^{-1}$ are marked by tip

down triangles, squares, tip top triangles, circles, diamonds, and pentagons, respectively. The dashed lines and open symbols indicate the single particle continuum regime. The insets show the dependency predicted by the coupled 2D plasmon model due to ref. 40. At $q_\perp = q_{||} = 0$ the curves diverge indicated by a dashed line.

was separately aligned with the help of the quasi-elastic peak. After alignment of the zero loss position a superposition of the following three functions was fitted to the spectra. (i) an asymmetric Pseudo-Voigt-profile $[V = c\left(\eta \frac{1}{1+(\omega-\omega_0)^2/(\lambda(\omega)\sigma)^2} + (1-\eta)\exp\left\{-\ln(2)\left(\frac{\omega-\omega_0}{\sigma}\right)^2\right\}\right)$ with $\lambda(\omega) = 1|_{\omega<=0}$ and $\lambda(\omega) > 1|_{\omega>0}]$ reflecting the quasi elastic peak including ultra-low-loss excitations such as phonons, (ii) a Drude-like function $I(\omega) = a\frac{\omega_P^2\omega\Gamma}{(\omega^2-\omega_P^2)^2+(\omega\Gamma)^2}$ corresponding to the plasmon peaks and (iii) a phenomenological background following a linear function. Finally, the spectral positions and half widths of the plasmons were derived from the fitted parameters $\omega_P$ and $\Gamma$ of the Drude function. The intensity (Fig. 7) corresponds to the area under the fitted Drude peak, which is obtained by integrating the latter over $\omega$.

### Calculation of the average Fermi velocity, susceptibility, and loss function

For the calculation of the Fermi velocity along the [100] direction, the bare particle susceptibility $\chi_0(q_{||}, \omega)$ and the loss function we used a TB

band structure[49] based on an LDA band calculation. $\chi_0(q_{||}, \omega)$ is calculated from a multi-band version of Eq. (2), taking only intra-band excitations with the same matrix element into account, thus neglecting inter-band excitations which may be caused by a strong spin-orbit coupling, which leads to a $k$-dependence of the orbital character of the bands[64]. Mainly the XY and the YZ bands contribute to $\langle v_F^2\rangle_{100}$ and $\chi_0$. The dielectric function and the loss function were calculated from Eqs. (1), (2), and (4). $\epsilon_b$ and the matrix element were fixed by using the plasmon energy from optical spectroscopy. For the calculation we used a half width of $\Gamma = 0.1$ eV.

### Data availability

All data supporting the findings are provided as figures in the article. Data files for all figures are available at https://opara.zih.tu-dresden.de/handle/123456789/1382 and from the corresponding authors on request.

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

**Table 1 | Acquisition times (in sec) for the used momentum transfers (in Å$^{-1}$)**

| $q_\perp$ <br> $q_{||}$ | 0.08 | 0.12 | 0.16 | 0.33 | 0.49 | 0.66 |
|---|---|---|---|---|---|---|
| 0 | 2 | 5 | 5 | 5 | 10 | 10 |
| 0.1 | 5 | 5 | 5 | 5 | 10 | 10 |
| 0.1 | 5 | 5 | 5 | 5 | 10 | 10 |
| 0.3 | 10 | 10 | 10 | 10 | 10 | 10 |
| 0.4 | 10 | 10 | 10 | 10 | 10 | 10. |

6. Norman, M. R. & Pépin, C. The electronic nature of high temperature cuprate superconductors. *Rep. Prog. in Phys.* **66**, 1547 (2003).

7. Lindhard, J. Kgl. Danske Videnskab. Selskab, Mat-phs. Medd **28**, 8 (1954).

8. Raether, H. *Excitation Of Plasmons And Interband Transitions By Electrons* 1st edn, Vol. 198 (Springer Verlag, Berlin, 1980).

9. Schnatterly, S. Inelastic electron scattering spectroscopy. *Solid State Phy.* **34**, 275–358 (1979).

10. Fink, J. Recent developments in energy-loss spectroscopy. *Adv. Electron Phys.* **75**, 121–232 (1989).

11. Nücker, N. et al. Plasmons and interband transitions in $Bi_2Sr_2CaCu_2O_8$. *Phys. Rev. B* **39**, 12379 (1989).

12. Wang, Y.-Y., Feng, G. & Ritter, A. L. Electron-energy-loss and optical-transmittance investigation of $Bi_2Sr_2CaCu_2O_8$. *Phys. Rev. B* **42**, 420 (1990).

13. Romberg, H. et al. Dielectric function of $YBa_2Cu_3O_{7-\delta}$ between 50 meV and 50 eV. *Zeitschrift für Physik B Condensed Matter.* **78**, 367 (1990).

14. Nücker, N., Eckern, U., Fink, J. & Müller, P. Long-wavelength collective excitations of charge carriers in high-$T_c$ superconductors. *Phys. Rev. B* **44**, 7155 (1991).

15. Knupfer, M., Roth, G., Fink, J., Karpinski, J. & Kaldis, E. Plasmon dispersion and the dielectric function in $YBa_2Cu_4O_8$ single crystals. *Phys. C Supercond.* **230**, 121 (1994).

16. Roth, F., Revcolevschi, A., Büchner, B., Knupfer, M. & Fink, J. Evidence for an orbital dependent Mott transition in the ladders of $(La, Ca)_xSr_{14-x}Cu_{24}O_{41}$ derived by electron energy loss spectroscopy. *Phys. Rev. B* **101**, 195132 (2020).

17. Terauchi, M. et al. Electron-energy-loss spectroscopy of oxide superconductor $Bi_2Sr_2CaCu_2O_8$. *Jpn. J. Appl. Phys.* **34**, L1524 (1995).

18. Terauchi, M., Tanaka, M., Tsuno, K. & Ishida, M. Development of a high energy resolution electron energy-loss spectroscopy microscope. *J. Microsc.* **194**, 203–209 (1999).

19. Romero-Bermúdez, A., Krikun, A., Schalm, K. & Zaanen, J. Anomalous attenuation of plasmons in strange metals and holography. *Phys. Rev. B* **99**, 235149 (2019).

20. Van den Eede, S. T., van Stralen, T. J. N., Flipse, C. F. J. & Stoof, H. T. C. Plasmons in a layered strange metal using the gauge-gravity duality. *Phys. Rev. B* **109**, 085119 (2024).

21. Mitrano, M. et al. Anomalous density fluctuations in a strange metal. *Proc. Natl. Acad. Sci. USA* **115**, 5392 (2018).

22. Husain, A. A. et al. Crossover of charge fluctuations across the strange metal phase diagram. *Phys. Rev. X* **9**, 041062 (2019).

23. Husain, A. A. et al. Pines' demon observed as a 3D acoustic plasmon in $Sr_2RuO_4$. *Nature* **621**, 66–70 (2023).

24. Chen, J. et al. Consistency between reflection momentum-resolved electron energy loss spectroscopy and optical spectroscopy measurements of the long-wavelength density response of $Bi_2Sr_2CaCu_2O_{8+x}$. *Phys. Rev. B* **109**, 045108 (2024).

25. Mauri, E., Smit, S., Golden, M. S. & Stoof, H. T. C. Gauge-gravity duality comes to the laboratory: Evidence of momentum-dependent scaling exponents in the nodal electron self-energy of cuprate strange metals. *Phys. Rev. B* **109**, 155140 (2024).

26. Grigoryan, V. G., Paasch, G. & Drechsler, S.-L. Determination of an effective one-electron spectrum from the plasmon dispersion of nearly optimally doped $Bi_2Sr_2CaCu_2O_8$. *Phys. Rev. B* **60**, 1340 (1999).

27. vom Felde, A., Sprösser-Prou, J. & Fink, J. Valence-electron excitations in the alkali metals. *Phys. Rev. B* **40**, 10181 (1989).

28. Fleszar, A., Stumpf, R. & Eguiluz, A. G. One-electron excitations, correlation effects, and the plasmon in cesium metal. *Phys. Rev. B* **55**, 2068 (1997).

29. Damascelli, A. et al. Fermi surface, surface states, and surface reconstruction in $Sr_2RuO_4$. *Phys. Rev. Lett.* **85**, 5194 (2000).

30. Chandrasekaran, A. et al. On the engineering of higher-order van Hove singularities in two dimensions. *Nat. Commun.* **15**, 9521 (2024).

31. Hepting, M. et al. Three-dimensional collective charge excitations in electron-doped copper oxide superconductors. *Nature* **563**, 374 (2018).

32. Nag, A. et al. Detection of acoustic plasmons in hole-doped lanthanum and bismuth cuprate superconductors using resonant inelastic X-Ray scattering. *Phys. Rev. Lett.* **125**, 257002 (2020).

33. Singh, A. et al. Acoustic plasmons and conducting carriers in hole-doped cuprate superconductors. *Phys. Rev. B* **105**, 235105 (2022).

34. Maeno, Y. et al. Superconductivity in a layered perovskite without copper. *Nature* **372**, 532 (1994).

35. Bergemann, C., Mackenzie, A. P., Julian, S. R., Forsythe, D. & Ohmichi, E. Quasi-two-dimensional Fermi liquid properties of the unconventional superconductor $Sr_2RuO_4$. *Adv. Phys.* **52**, 639 (2003).

36. Mackenzie, A. P. et al. Extremely strong dependence of superconductivity on disorder in $Sr_2RuO_4$. *Phys. Rev. Lett.* **80**, 161 (1998).

37. de' Medici, L., Mravlje, J. & Georges, A. Janus-faced influence of Hund's rule coupling in strongly correlated materials. *Phys. Rev. Lett.* **107**, 256401 (2011).

38. Knupfer, M., Jerzembeck, F., Kikugawa, N., Roth, F. & Fink, J. Propagating charge carrier plasmons in $Sr_2RuO_4$. *Phys. Rev. B* **106**, L241103 (2022).

39. Grecu, D. Plasma frequency of the electron gas in layered structures. *Phys. Rev. B* **8**, 1958 (1973).

40. Fetter, A. L. Electrodynamics of a layered electron gas. II. Periodic array. *Ann. Phys.* **88**, 1 (1974).

41. Das Sarma, S. & Quinn, J. J. Collective excitations in semiconductor superlattices. *Phys. Rev. B* **25**, 7603 (1982).

42. Greco, A., Yamase, H. & Bejas, M. Plasmon excitations in layered high-$T_c$ cuprates. *Phys. Rev. B* **94**, 075139 (2016).

43. Abbamonte, P. & Fink, J. Collective charge excitations studied by electron energy-loss spectroscopy. *Ann. Rev.Condensed Matter Phys.* **16**, 465 (2025).

44. Platzman, P. M. & Wolff, P. A. *Waves And Interactions In Solid State Plasmas* Vol. 304 (Academic Press, New York, 1973).

45. Ehrenreich, H. & Cohen, M. H. Self-consistent field approach to the many-electron problem. *Phys. Rev.* **115**, 786 (1959).

46. Hepting, M. et al. Gapped collective charge excitations and inter-layer hopping in cuprate superconductors. *Phys. Rev. Lett.* **129**, 047001 (2022).

47. Apostol, M. Plasma frequency of the electron gas in layered structures. *Z. für Phys. B Condens. Matter* **22**, 13 (1975).

48. Stricker, D. et al. Optical response of $Sr_2RuO_4$ reveals universal fermi-liquid scaling and quasiparticles beyond Landau theory. *Phys. Rev. Lett.* **113**, 087404 (2014).

49. Liebsch, A. & Lichtenstein, A. Photoemission quasiparticle spectra of $Sr_2RuO_4$. *Phys. Rev. Lett.* **84**, 1591 (2000).

50. Batson, P. & Silcox, J. Experimental energy-loss function, $Im[-1/\varepsilon(q,\omega)]$, for aluminum. Phys. Rev. B 27, 5224 (1983).

51. Engelsberg, S. & Schrieffer, J. R. Coupled electron-phonon system. *Phys. Rev.* **131**, 993 (1963).

52. Ashcroft, N. W. & Mermin, N. D. *Solid State Physics,* Vol. 848 (Saunders College, 1976).

53. Nag, A. et al. Impact of electron correlations on two-particle charge response in electron- and hole-doped cuprates. *Phys. Rev. Res.* **6**, 043184 (2024).

54. Deng, X. et al. How bad metals turn good: Spectroscopic signatures of resilient quasiparticles. *Phys. Rev. Lett.* **110**, 086401 (2013).

55. DuBois, D. F. & Kivelson, M. G. Electron correlational effects on plasmon damping and ultraviolet absorption in metals. *Physical Review* **186**, 409 (1969).

56. Hunter, A. et al. Fate of quasiparticles at high temperature in the correlated metal $Sr_2RuO_4$. *Phys. Rev. Lett.* **131**, 236502 (2023).

57. Sturm, K. Electron energy loss in simple metals and semi-conductors. *Adv. Phys.* **31**, 1 (1982).

58. Paasch, G. Influence of interband transitions on plasmons in the alkali metals: pseudopotential calculation. *Phys. stat. sol.* **38**, K123 (1970).

59. Gibbons, P. C. & Schnatterly, S. E. Comment on the line shape of the plasma resonance in simple metals. *Phys. Rev. B* **15**, 2420 (1977).

60. Sturm, K. & Gusarov, A. Dynamical correlations in the electron gas. *Phys. Rev. B* **62**, 16474 (2000).

61. Mahan, G. D. *Many-Particle Physics*, Third Edition Vol. 788 (Plenum, New York, 2000).

62. Khaliullin, G. & Horsch, P. Theory of the density fluctuation spectrum of strongly correlated electrons. *Phys. Rev. B* **54**, R9600 (1996).

63. Bobowski, J. S. et al. Improved single-crystal growth of $Sr_2RuO_4$. *Condens. Matter* **4**, 6 (2019).

64. Tamai, A. et al. High-resolution photoemission on $Sr_2RuO_4$ reveals correlation-enhanced effective spin-orbit coupling and dominantly local self-energies. *Phys. Rev. X* **9**, 021048 (2019).

## Acknowledgements

J.F. thanks P. Abbamonte, R. von Baltz, S.-L. Drechsler, and A. Greco for helpful discussions. This work is supported by a KAKENHI Grants-in-Aids for Scientific Research (Grants No. 18K04715, No. 21H01033, and No. 22K19093), and Core-to-Core Program (No. JPJSCCA20170002) from the Japan Society for the Promotion of Science (JSPS) and by a JST-Mirai Program (Grant No. JPMJMI18A3). J.S. received funding from the HORIZON EUROPE framework program for research and innovation under grant agreement n. 101094299. A.L. and M.K. acknowledge funding from the Deutsche Forschungsgemeinschaft (DFG, German Research Foundation)-project-id 461150024. B.B. received funding from the Würzburg-Dresden Cluster of Excellence on Complexity and Topology in Quantum Matter - ct.qmat (EXC 2147, project-id 390858490).

## Author contributions

J.F., M.K., A.L., and B.B. conceived the experiment, J.S. and D.W. performed the EELS experiment. J.S. and J.F. analyzed the data. F.J. and N.K. prepared and characterized the samples. J.F., J.S., and A.L. calculated the susceptibility and the loss function. J.F., J.S., and A.L. wrote the manuscript with input from all authors.

## Funding

## Competing interests

The authors declare no competing interests.
