## [Transparent Peer Review file · Nature Communications]

Optical and acoustic plasmons in the layered material Sr_2RuO_4

Corresponding Author: Dr Johannes Schultz

Version 0:

Reviewer comments:

Reviewer #1

(Remarks to the Author)

The manuscript “Optical and acoustic plasmons in the layered material Sr_2RuO_4 ” by J. Schultz and coworkers reports the observation of plasmons in strontium ruthenate with high-resolution EELS. The authors compare the experimental data with RPA theoretical calculations and conclude that the collective mode propagation is explained in terms of unrenormalized electron motion. This is an important topic of widespread interest, as the rapid progress in holographic and other theories of strange metals and related materials have raised the urgency of measurements of the density fluctuations. Taken at face value, the data are beautiful and worth publishing in some form.

However, I have serious reservations about the claims made by the authors, and whether they are supported by the data. The main issue I find is that the claim of consistency with unrenormalized electron mass is at odds with optics and ARPES measurements of this compound. Further, the manuscript contains several inaccuracies that raise concerns about the solidity of the data interpretation. Therefore, in my opinion the paper is not publishable at all in its current form in Nature Communications or in any other venue. Specifically, the authors need to address the points described below.

Comments on the Introduction:

1. The current abstract and introduction do not ask the scientific question that these measurements are trying to answer. The reader is provided with an introduction in medias res with a generic reference to strange metals, which is unsuitable for the broad readership of Nat Commun. Sr_2RuO_4 has been recently investigated with RIXS and EELS by other groups as it is a clean material with a clear crossover between Fermi liquid and non-Fermi liquid behavior [PRL 113, 087404 (2014)]. In essence, Sr_2RuO_4 can be seen as a fruit fly for the emergence of strange-metal behavior at high temperature. The introduction needs to be rewritten to better formulate the scientific case and eliminate the impression that this work simply parrots other recent contributions to the research area.
2. The authors state that Sr_2RuO_4 is a “strange” metal, but in fact this material is an excellent Fermi liquid with $\rho \sim T^2$ resistivity, nice quantum oscillations, etc [Advances in Physics, 2003, 52, 639–725]. It only has strange metal like properties on high energy scales, $\omega > 0.1$ eV or higher. I recommend the authors to be more precise about this distinction.
3. The authors claim that acoustic plasmons detected with RIXS are underdamped. But in fact these are overdamped and have a width that is comparable to what was observed in reflection EELS [see PRL 125, 257002 (2020), Figs. S6-S11 of the supplementary information]. The authors need to correct this statement to ensure consistency with the x-ray data.
4. Note that plasmons in cuprates are not thought to be acoustic, as interlayer hopping makes the Fetter model invalid at $q_z = \pi$. The authors need to revise the manuscript in order to include interlayer hopping effects.
5. Authors should not use “acoustic” and “optic” as terms for the plasmon in the Fetter model. There is just one plasmon branch whose frequency depends on both q and q_z , and vanishes as $q \rightarrow 0$ for any nonzero value of q_z . There are no distinct branches like the acoustic and optical phonon modes in a crystal. The authors need to correct their language about this.
6. The introduction, the authors cite T-EELS papers by Nucker and Wang claiming a dispersing plasmon in $\text{Bi}_2\text{Te}_2\text{Se}$. But T-

EELS measurements by Terauchi et al. in Sendai, performed around the same time, claim there is no plasmon at all in Bi2212 at a $q \sim 0.3 \text{ \AA}^{-1}$, and see a continuum like what was observed in reflection EELS measurements [Journal of Microscopy, 194, 203–209 (1999); Jpn. J. Appl. Phys. 34 L1524 (1995)]. The authors should cite these papers in context to provide a more comprehensive picture of the field.

7. The review of experimental R-EELS efforts should include the latest R-EELS paper, Chen et al., PRB, 109, 045108 (2024).

Comments on the body of the paper:

1. The authors core claim, that the plasmon in Sr2RuO4 can be explained by noninteracting electrons, is not supported by their data. The dielectric function measured with EELS should be the same as that measured by optics at $q=0$. The claim that the electron motion is compatible with an unrenormalized electron mass is incompatible with the finding that $m^*/m \sim 3$ in Sr2RuO4, and $m^*/m \sim 4$ in LSCO and LBCO [PRB 72, 060511R (2005)]. Optical studies of Sr2RuO4 [PRL 113, 087404 (2014)] clearly show extra weight above $\omega=0.1 \text{ eV}$ that is a sign of Hund's rule exchange and requires dynamical mean field theory (DMFT) to explain. The information about Hund's rule/DMFT effects are contained in the lineshape of the plasmon, not its energy. The center dispersion of the plasmon is nonspecific; it is just determined by geometry and the long-ranged Coulomb interaction. It should not differ appreciably between strong and weak coupling scenarios, which are reflected in the plasmon lineshape. The authors either need to do a detailed lineshape analysis, of the sort done by Stricker et al., or modify their conclusions to match the information contained in their presentation of the data.

2. The authors need to show all of their data, including the elastic lines and their background, and explain how they processed and fit them. Methods Section B says the elastic peaks were subtracted from the data but the authors do not show the elastic peaks or give details about how this subtraction was done. The reason this is a concern is that, at $q_c=0$, the curves in Fig 3 (panels 1-6) show no dispersion at all, but a clear dispersion is shown in the summary points in Fig. 4. These two plots contradict one another, raising concerns about their data processing. The authors should therefore show their background, how they subtracted it, how the data look before and after the subtraction, and explain and justify the fit model that was used. They also need to quantify the resulting error, which occur with any model fit, and reflect this in the error bars on Fig. 4. This is standard scientific practice; it is surprising not to see basic things like quantifying sources of error in a paper by such a prominent group of authors.

3. The authors' RPA calculations contradict the RPA calculations presented in Husain et al. Nature 621, 66 (2023). In particular, that calculation shows negative plasmon dispersion, which is expected in presence of electronic correlations of the type present in Sr2RuO4. The authors need to explain why they get a different result, and justify whether and why their approach is more appropriate or correct than the approach in Husain et al.

4. The authors claim that the plasmon is Landau damped at large q . However, looking at the $q_z=0$ data (Fig. 3), the plasmon has the same width at $q=0.66$ as it does at $q=0.08$. The authors need to explain what the basis is for the claim of Landau damping when this is clearly not reflected in the width of the peak. I recommend adding another panel to Fig. 4 showing how the width of the peak (not just its energy) depends on q , and then discussing Landau damping in terms of this width.

5. What is the difference between χ and χ_{sc} in Eq. 3? In Nozieres and Pines, the screened χ_{sc} is the same as the Lindhard function, which is called χ_0 in this paper. Please clarify the notation.

6. I cannot understand the meaning of this sentence: "The rapid decay in the theoretical curve for the optical plasmon near $q_c = 0$ is due to the finite momentum resolution, i.e., integration of optical and acoustic plasmons within the EEL collection aperture." Why does the experimental resolution affect the theoretical curve? Please rephrase or clarify.

Reviewer #2

(Remarks to the Author)

The paper by Schultz et al. reports T-EELS experiments on Sr2RuO4 where the plasmon dispersion curves were measured. The authors report off axis measurement at various q values.

The Sr2RuO4 dispersion curves were measured in the (ac) plane by orienting the FIB prepared lamella with the e-beam parallel to the b-axis.

Although the study is well constructed and data supports the findings, I find that the impact of this paper might be lacking.

The data at 0.3 1/\AA seems to deviate strongly from the calculated curves, could the authors comment on this?

The authors report the energy of the plasmons for the different q values, but it seems to me that the intensity variation might also be interesting to plot as a function of q_c (maybe also as a function of q_a)

The first sentence of paragraph V seems to be lacking an end "not only"... should be followed by a but... or something similar

The impact of the measurements on the understanding of the correlation between acoustic, optical plasmons and superconductivity would be greatly improved by some more approachable schematic representation to guide the reader through this complex concept.

Similarly a more visual description of the correlation between the crystal structure anisotropy (layered structure with the acoustic/optical plasmon velocity) would also help into making the work accessible to a larger community.

I am missing a value of the collection angle and the acquisition time for the EELS experiments (the collection angle should give an idea of the q space resolution but this is not mentioned and might be difficult to follow for the non-expert reader)

Reviewer #3

(Remarks to the Author)

The authors perform a momentum-resolved transmission EELS measurement to investigate plasmons in the material Sr₂RuO₄. To my knowledge this is the most thorough study of this type on Sr₂RuO₄ to date. I believe the findings are valuable to the broad condensed matter community investigating correlated electron systems and strange metals in particular. The results should probably be published though I list below a few minor points and questions that could be clarified.

1. Should mass in Eq (4) be "effective mass"?
2. In Fig.4, why does the purple solid line ($q_c=0$) dip near $q=0$? Should it not approach $\omega_P(0)$ smoothly? (See Fig 1a of Ref. 2 for example.)
3. Eq (5) deserves either more details, such as a derivation, or a reference. Is it not obvious how to define "average squared Fermi velocity along the q direction"?
4. The left side of Eq (5) gives ω_P in the limit of small q . For the range of q in the experiment, have the authors checked whether the expression is applicable? For example the authors could calculate Eq (3) and check for what range of q do the calculated peak positions agree with the expression in Eq (5).
5. Was any data collected along the 110 direction?

Version 1:

Reviewer comments:

Reviewer #1

(Remarks to the Author)

The authors have provided a revised version of their manuscript that addresses fundamental structural flaws in the narrative. I commend them for documenting their data analysis and experimental methods in a way that conforms with the reporting standards of modern scientific literature. However, I find their reply and the increased focus on the lack of mass renormalization especially troubling.

The central claim of this paper is that "it is possible to describe the range of optical plasmon excitations at high energies in a mean-field random phase approximation without taking correlation effects into account." In their rebuttal, the authors claim that optics and ARPES supports their conclusions. Without commenting on the fact that the authors are conflating a discussion about the properties of cuprate plasmons with a study of the charge dynamics of Sr₂RuO₄, I will explain below why this conclusion is incorrect and inconsistent not only with a large body of literature, but also with their own data.

Optical data on cuprate superconductors by Basov, Van der Marel, and several other groups

(<https://journals.aps.org/prl/abstract/10.1103/PhysRevLett.95.097002>,

<https://journals.aps.org/rmp/abstract/10.1103/RevModPhys.83.471>

), show that the experimental optical effective mass is frequency-dependent but ultimately involves an integral over a wider range of energies. Further, as the authors certainly know, the relationship between optical conductivity and loss function is such that low-energy effects in the optical conductivity translate into high-energy effects in the loss function. As an example, the plasma frequency is defined by an integral of the optical conductivity from zero frequency to infinity, and the largest spectral weight comes from low frequencies in σ_1 . Stating that low-energy effects in the optical response are irrelevant for the loss function at the eV scale ignores this elementary connection between response functions. Optical data universally find signs of mass renormalization at low energy and even if at high energy the quasiparticles are not renormalized, their mass enhancement should be reflected at the 1.5 eV scale of the plasmon.

Even without resorting to these considerations, the authors' conclusions are ***inconsistent with their own data.*** If the charge dynamics is truly non-interacting, so much so that it can be described by RPA without mass renormalization, where does the width of the plasmon at $q=0$ come from? The answer is likely correlation effects, which the authors are deeming irrelevant. Fig. 5 extrapolates at $q \rightarrow 0$ to a 1 eV plasmon width for a plasma frequency of 1.5 eV. This is a sizeable damping and the authors articulate an explanation of their data that fails to account for their own experimental finding. I also note that the RPA theory grossly deviates from the experimental data in Fig. 4 for $q > 0.3 \text{ \AA}^{-1}$, meaning that this description is incomplete at best, wrong at worst.

The referee also notes that there is a successful RPA description of the charge spectrum of this material in Nature 621, 66–70 (2023). The authors explain that the difference between the two descriptions lies purely in the mass renormalization. However, the two pictures cannot both be correct, as the electrons cannot simultaneously have a m^*/m ratio of 4 and 1. How this discrepancy can be reconciled is not clear at this stage.

Finally, I find it striking that some of the authors here advocate for a lack of mass renormalization while in <https://arxiv.org/pdf/2407.15692> they invoke mass renormalization to describe plasmon dispersion and width. I quote here some excerpts of that manuscript, although related to cuprates and only loosely related to the plasmons of Sr₂RuO₄:

“We demonstrate that an appropriate fit to experimentally observed plasmon dispersion is possible within a RPA model with the inclusion of a system dependent band renormalization parameter, and without which, unrealistic values of dielectric constants and incoherent excitations are obtained.”

“to explain the experimental results in this approximate model, one needs to use renormalized band dispersions which amount to mass enhancement of $m^*/m = 2.0$ and $m^*/m = 1.7$ for LSCO and LCCO, respectively.”

“The inclusion of self-energy effects leads to a broadening of the plasmons (in addition to mass enhancement), and an energy dependence of the mass enhancement cannot be ruled out”

The meaning of these words is self-evident and in conflict with the key tenet of the manuscript under consideration. These statements confirm that mass renormalization effects are important to capture the plasmon dispersion in correlated materials.

Since the key thesis of the work is questionable and at odds with decades of ARPES, optics, RIXS, as well as in clear contrast with the authors' data, I cannot endorse this paper for publication.

Reviewer #2

(Remarks to the Author)

The authors have answered most of my concerns so I recommend publication of this manuscript.

Reviewer #3

(Remarks to the Author)

I have reviewed the changes to the manuscript made by the authors. All points I have raised in the previous round have been addressed satisfactorily. I am also pleased to see that the authors have substantially expanded their introduction according to the suggestions of Referee 1. I recommend the manuscript for publication in Nature Comm.

Version 2:

Reviewer comments:

Reviewer #1

(Remarks to the Author)

The authors have provided a revised version of their manuscript and a reply to my previous report. As I previously wrote, the claim of lack of mass renormalization is troubling and at odds with the authors' data. While I will elaborate below on each point raised by the authors, I will immediately state that the paper cannot be accepted in its present form. The experimental data contained in this work is valuable, but for this work to belong to the scientific literature, I advise the editor to recommend the authors to drop the claim of unrenormalized electronic masses.

Regarding their rebuttal, the authors argue in point (1) against my statement that the mass renormalization found at low energy will affect the plasmon dispersion. They do so by producing a new figure 5 including RPA calculations for three model effective masses of the carriers. They find that introducing a smooth dependence from a renormalized mass of 3.5 to an unrenormalized mass at high energy yields negligible changes of the plasmon dispersion compared with a fully unrenormalized mass. This argument is clever but lacks a physical explanation to ground what otherwise remains an ad hoc feature to reproduce the data. In this sense the authors' logic is circular. In addition, Fig. 5 makes clear that their plasmon dispersion from Fig. 4 is much more similar to the one they obtain for $m^*/m=3.5$ (minus the q_z dependence from the layered nature of the electron gas). I would encourage the author to plot the two dispersions on top of the data and see which fits best. It is unclear to this reviewer how the authors can reconcile the dispersion found in Fig. 4 with their core claim. However, the magnitude of this discrepancy in the economy of the claim and with rest of the optics/ARPES/RIXS literature is too big to ignore.

Regarding point 2, the reference to interband transitions is interesting, but needs to be substantiated. The authors write in the paper that “Already in our previous paper [36], we have emphasised that the width is almost constant in the studied low momentum range. This means that it is not related to electron-electron interaction which would lead to a quadratic increase of the width as a function of momentum transfer [50].” The momentum-dependent width in Fig. 6 looks quadratic to me, and I encourage the authors to deepen the discussion of this issue if they want to truly settle the issue correlations vs interband transitions. At this stage, the paper says too little to address the matter and the arguments in favor of interband transitions must be taken at authors' word.

Point 3 does not yet address the fundamental discrepancy between experiment and unrenormalized mass RPA, in the sense that the RPA provided by the authors in Fig. 5 convinced me even more that the mass is renormalized, as they see a dispersion in Fig. 4 akin to the unlabeled panel in second row, third column of Fig. 5.

Regarding point 4 of the rebuttal, I just note that, in addition to the irreducible incompatibility with the RPA in Husain et al, the RPA of this manuscript is also incompatible with calculations recently done by the Cohen group PRB 110, 155127 (2024).

Finally, concerning point 5, I will leave to the editor and the readers to judge if it is a sustainable position to advocate for a lack of mass renormalization here while invoking mass renormalization to describe plasmon dispersion and width in <https://arxiv.org/pdf/2407.15692> at the same time. The authors suggest I have taken sentences out of context from their other work. However, I will note that even the abstract of that work states "The t-J -V model, which includes the correlation effects implicitly, accurately describes the plasmon dispersions as resonant excitations outside the single-particle intra-band continuum. In comparison, a quantitative description of the plasmon dispersion in the RPA approach is obtained only upon explicit consideration of re-normalized electronic band parameters. Our comparative analysis shows that electron correlations significantly impact the low-energy plasmon excitations across the cuprate doping phase diagram, even at long wavelengths. Thus, complementary information on the evolution of electron correlations, influenced by the rich electronic phases in condensed matter systems, can be extracted through the study of two-particle charge response." The claim contained even in the abstract of that work is in conflict with the message of the paper under consideration here. The authors try to explain this inconsistency away by talking about acoustic plasmons, but, by using their own words, the Sr₂RuO₄ plasmons at selected q_z are also acoustic, hence squarely within the scope of the other work. I maintain that the key thesis of the present manuscript is questionable and at odds with decades of ARPES, optics, RIXS, as well as in clear contrast with the authors' data.

Reviewer #4

(Remarks to the Author)

I find the work quite interesting as it reports a thorough study of collective excitations, plasmons, in the layered material Sr₂RuO₄. This quantum material can be considered as a benchmark material to be understood as it displays excellent Fermi liquid behaviour at low temperatures, "strange metal" behaviour at high temperatures, superconductivity, Van Hove singularities. Therefore the results are very timely and rather conclusive. I also find the conclusion that there are no signs of over-damped plasmons as predicted by holographic theories, rather important.

As far as I can see, the most important questions, which are (i) the mass renormalisation and (ii) the interaction effects (beyond RPA) has been raised by the previous referees and have been addressed in a satisfactory and quite detailed manner.

One point that it is worth making, given the wide interest of the material, is to connect and comment on the results with the view that the surface of Sr₂RuO₄ behaves differently, due to energetically favourable rotation of octahedra (see for example Nature Communications 15 (1), 9521 (2024) and references therein). A simple argument based on the nature of the experiments will be enough to make the distinction, and I believe it will be informative to the readers.

After that remark, I am happy to recommend the manuscript for publication in Nature Communications as it is a study of high standards and significance.

Version 3:

Reviewer comments:

Reviewer #4

(Remarks to the Author)

The authors have convincingly replied to all comments and questions. One note that may facilitate the argument regarding the effective mass is that it is entirely physical for the effective mass to energy (and sometimes momentum) dependent. I recommend publication in Nature Communication.

Reviewer Comments and Author Rebuttals

Reviewer 1

The manuscript Optical and acoustic plasmons in the layered material Sr_2RuO_4 by J. Schultz and coworkers reports the observation of plasmons in strontium ruthenate with high-resolution EELS. The authors compare the experimental data with RPA theoretical calculations and conclude that the collective mode propagation is explained in terms of unrenormalized electron motion. This is an important topic of widespread interest, as the rapid progress in holographic and other theories of strange metals and related materials have raised the urgency of measurements of the density fluctuations. Taken at face value, the data are beautiful and worth publishing in some form.

We thank the Referee for the statements, that the data are beautiful, that the paper deals with an important topic, and worth publishing in some form. In addition, we mention that to our knowledge, there are no RIXS results of acoustic plasmon excitations in Sr_2RuO_4 . We only found a RIXS study of spin-orbit splitting of the t_{2g} Ru states in the literature [1].

However, I have serious reservations about the claims made by the authors, and whether they are supported by the data. The main issue I find is that the claim of consistency with unrenormalized electron mass is at odds with optics and ARPES measurements of this compound. Further, the manuscript contains several inaccuracies that raise concerns about the solidity of the data interpretation. Therefore, in my opinion the paper is not publishable at all in its current form in Nature Communications or in any other venue. Specifically, the authors need to address the points described below.

The main concern of Reviewer 1 is that we explain our EELS results with an unrenormalized electron mass. Yes, this is one of our main points and it is in contradiction with holographic theories for strange metals and with data using R-EELS [2, 3, 4] on cuprates and Sr_2RuO_4 . This led Reviewer 1 to the statement that our paper is not publishable at all in Nature Communication or in any other venue. Different from Reviewer 1, we think optics [5] and ARPES [6, 7] do even support our interpretation. Both methods observe an energy dependence of the effective mass which would lead at 1.5 eV (the optical plasmon energy) to an unrenormalized band structure. In the revised paper, we further corroborated this result by various additional analysis (e.g., plasmon width and intensity). We therefore strongly disagree with the statement that the paper is not publishable at all in its current form in Nature Communications or in any other venue.

1. The current abstract and introduction do not ask the scientific question that these measurements are trying to answer. The reader is provided with an introduction in medias res with a generic reference to strange metals, which is unsuitable for the broad readership of Nat Commun. Sr_2RuO_4 has been recently investigated with RIXS and EELS by other groups as it is a clean material with a clear crossover between Fermi liquid and non-Fermi liquid behavior. [PRL 113, 087404 (2014)]. In essence, Sr_2RuO_4 can be seen as a fruit fly for the emergence of strange-metal behavior at high tempera-

ture. The introduction needs to be rewritten to better formulate the scientific case and eliminate the impression that this work simply parrots other recent contributions to the research area.

We followed the advice of Reviewer 1 and have rewritten the Introduction. We hope that it better describes the scientific case.

2. The authors state that Sr_2RuO_4 is a strange metal, but in fact this material is an excellent Fermi liquid with $\rho \propto T^2$ resistivity, nice quantum oscillations, etc [Advances in Physics, 2003, 52, 639725]. It only has strange metal like properties on high energy scales, $\omega > 0.1$ eV or higher. I recommend the authors to be more precise about this distinction.

We followed the advice of Reviewer 1 and have rewritten the Introduction. We think that now, we have better described the temperature dependence of the electronic structure of Sr_2RuO_4 .

3. The authors claim that acoustic plasmons detected with RIXS are underdamped. But in fact these are overdamped and have a width that is comparable to what was observed in reflection EELS [see PRL 125, 257002 (2020), Figs. S6- S11 of the supplementary information]. The authors need to correct this statement to ensure consistency with the x-ray data.

We agree with the Reviewer, that there is some overdamping in the RIXS spectra of cuprates [8]. On the other hand, the authors of Ref. [8] state: "Since Γ influences the width of the plasmon, its effect on the plasmon peak position is strongest when it becomes comparable to undamped plasmon energy (overdamped condition). As observed in the experiment (see Fig. S12 (d, e) and (f)), this condition may only be true close to the zone-centre for the acoustic plasmons." They also state that the width is composed by an intrinsic broadening due to correlation effects and an external effect due to a finite resolution. Thus overdamping in the centre of the BZ may occur due to a finite resolution. We have added a remark that overdamping only occurs close to the zone center.

4. Note that plasmons in cuprates are not thought to be acoustic, as interlayer hopping makes the Fetter model invalid at $q_z = \pi$. The authors need to revise the manuscript in order to include interlayer hopping effects.

We agree with the Reviewer. For each set of q_{\parallel}, q_{\perp} there is a plasmon branch which changes its character: for $q_{\perp} = 0$ there is an "optical" plasmon with a gap, which decays for $q > q_{\text{crit}}$ due to Landau damping, i.e., due to a decay into particle-hole excitations from the continuum. With increasing q_{\perp} the plasmon moves at low momentum to lower energy forming a gap-less "acoustic" plasmon with a linear in energy dispersion. For a finite inter layer hopping a gap appears at low energy and momentum. We have changed the text but we apply the terms which are used in the literature.

5. Authors should not use acoustic and optic as terms for the plasmon in the Fetter model. There is just one plasmon branch whose frequency depends on both q and q_z , and vanishes as $q \rightarrow 0$ for any nonzero value of q_z . There are no distinct branches like

the acoustic and optical phonon modes in a crystal. The authors need to correct their language about this.

In this case we do not follow the Reviewer. The complete EELS, RIXS, and theory community in this research field uses this terminology.

6. The introduction, the authors cite T-EELS papers by Nucker and Wang claiming a dispersing plasmon in Bi2212. But T-EELS measurements by Terauchi et al. in Sendai, performed around the same time, claim there is no plasmon at all in Bi2212 at a q 0.3 \AA^{-1} , and see a continuum like what was observed in reflection EELS measurements [Journal of Microscopy, 194, 203209 (1999); Jpn. J. Appl. Phys. 34 L1524 (1995)]. The authors should cite these papers in context to provide a more comprehensive picture of the field.

This discrepancy can be easily explained by the used momentum resolution in the experiments. Nuecker et al. used a momentum resolution of 0.04\AA^{-1} . Therefore, they could differentiate between coherent plasmon ($q < q_{crit}$) and incoherent particle-hole ($q > q_{crit}$) excitations. Terauchi et al. used an momentum resolution of 30\AA^{-1} . Thus they recorded mainly the incoherent continuum.

In the revised version we have cited the Sendai work.

7. The review of experimental R-EELS efforts should include the latest R-EELS paper, Chen et al., PRB, 109, 045108

In the revised version, we have cited the latest R-EELS paper, Chen et al., PRB, 109, 045108 (2024).

Comments on the body of the paper

1. The authors core claim, that the plasmon in Sr_2RuO_4 can be explained by noninteracting electrons, is not supported by their data. The dielectric function measured with EELS should be the same as that measured by optics at $q=0$. The claim that the electron motion is compatible with an unrenormalized electron mass is incompatible with the finding that $m^*/m \approx 3$ in Sr_2RuO_4 , and $m^*/m \approx 4$ in LSCO and LBCO PRB 72, 060511R (2005). Optical studies of Sr_2RuO_4 PRL 113, 087404 (2014) clearly show extra weight above $\omega=0.1$ eV that is a sign of Hund's rule exchange and requires dynamical mean field theory (DMFT) to explain. The information about Hund's rule/DMFT effects are contained in the lineshape of the plasmon, not its energy. The center dispersion of the plasmon is nonspecific; it is just determined by geometry and the long-ranged Coulomb interaction. It should not differ appreciably between strong and weak coupling scenarios, which are reflected in the plasmon lineshape. The authors either need to do a detailed lineshape analysis, of the sort done by Stricker et al., or modify their conclusions to match the information contained in their presentation of the data.

Different from Referee 1, we believe that the data on the plasmon dispersion can be explained by non-interacting electrons. We kindly point out, that the cited work [9] reports effective mass values at low energies. The effective mass data on LBCO and YBCO were derived from a combination of Hall effect data and optical data limiting the

energy at 650 cm^{-1} . The values of these data also agree with the low energy specific heat data. However, recent ARPES data [7] and optical data on Sr_2CuO_4 [5] show that at high temperatures/energies, the renormalisation function Z increases and therefore the effective mass decreases. For example, in Sr_2RuO_4 the optical effective mass decreases from ≈ 3.4 at low energies to ≈ 1.7 at 200 meV [5]. Because in cuprates ($\omega_P \approx 1 \text{ eV}$) and in Sr_2CuO_4 ($\omega_P \approx 1.5 \text{ eV}$) the optical plasmon energy is considerably higher than the Hund's exchange interaction ($J \approx 0.3 \text{ eV}$) the renormalization is only important at low energies/temperature. Such resilient quasiparticles were predicted also by DMFT calculations [5, 6, 7, 10]. A reduction of the effective mass at high energies has been described in the phenomenological marginal Fermi liquid model [11, 12] as well.

Concerning the plasmon lineshape, optical data of the loss function in *optimally* doped cuprates [9] yields a perfect Drude lineshape. Spectral weight in the MIR occurs only in the *underdoped* compounds. Already in our first T-EELS paper [13] on Sr_2CuO_4 we have presented data on the momentum dependence of the line width. Within error bars, the plasmon can be fitted within a Drude-Lorentz model up to q_{crit} . In the revised manuscript, we have added a figure in which we depict the momentum dependence of the plasmon width from our present data together with the data from [13].

Concerning the plasmon dispersion, we also disagree with the Referee 1. We believe that the plasmon dispersion is related to the compressibility of the electron liquid and the dynamic local field corrections of the electron liquid and therefore provides important information on the many-body physics of metals [14, 15, 16, 17]. In this manuscript we derive the important result, that many-body correlation effects caused by spin dynamics ($J \approx 0.3 \text{ eV}$) or any low-energy boson are important at low energies, but not at the plasmon energies of above 1 eV. Therefore, our T-EELS data of the optical plasmon dispersion can be explained with an unrenormalized band structure. These data also do not support overdamped plasmons which are predicted in terms of holographic theories. In this context, we mention that our calculations of the dispersion deliver a positive plasmon dispersion for an unrenormalized band structure, but a small or a negative dispersion for a renormalized band structure with an effective mass equal to four [13].

2. The authors need to show all of their data, including the elastic lines and their background, and explain how they processed and fit them. Methods Section B says the elastic peaks were subtracted from the data but the authors do not show the elastic peaks or give details about how this subtraction was done. The reason this is a concern is that, at $q_c=0$, the curves in Fig 3 (panels 1-6) show no dispersion at all, but a clear dispersion is shown in the summary points in Fig. 4. These two plots contradict one another, raising concerns about their data processing. The authors should therefore show their background, how they subtracted it, how the data look before and after the subtraction, and explain and justify the fit model that was used. They also need to quantify the resulting error, which occur with any model fit, and reflect this in the error bars on Fig. 4. This is standard scientific practice; it is surprising not to see basic things like quantifying sources of error in a paper by such a prominent group of authors.

In the revised version we optimized the data processing (including the elastic peak into the fit instead of subtraction), explain it in more detail, and show the untreated data as

well. However, we can not comprehend the comment concerning contradicting data for $q_c = 0$ in Figs. 3 and 4. In the panels 1-6 of Fig. 3 of the first version one can clearly see an increase of the plasmon energy. Starting from below 1.5 eV (panel 1) it increases to around 1.5 eV (panels 2 and 3) and further to clearly above 1.5 eV (panels 4 and 5). The data for $q_c = 0$ in Fig. 4 (purple squares) fit exactly to the behaviour obtained by estimating the spectral positions in Fig. 3 described above. The same holds for the data in Figs. 3 and 4 in the revised version of the paper (although the energies have changed slightly due to the improved data analysis). The reason for the concern is probably a misinterpretation of the purple diamonds in Fig. 4. As written in the caption they do not correspond to our data at $q_c = 0$. This points are plotted for comparison and correspond to the data derived in our previous publication [13] where only the dispersion of the optical plasmon was measured. To distinguish these data points better from the recent data, we changed the color (dark purple) and marker (stars) for this data points. Concerning the error bars: We included both, the error given by the limited spectral resolution of the EELS device as well as the fitting error. Since the latter is much smaller, the length of the error bar is dominated by the error of the device, which is constant for all data points. We added a comment on that in the caption of Fig. 4.

3. The authors RPA calculations contradict the RPA calculations presented in Husain et al. Nature 621, 66 (2023). In particular, that calculation shows negative plasmon dispersion, which is expected in presence of electronic correlations of the type present in Sr2RuO4. The authors need to explain why they get a different result, and justify whether and why their approach is more appropriate or correct than the approach in Husain et al.

The difference can be explained in the following way: Husain has used a band structure from ARPES data which is renormalized by a factor of about 4. We have used an unrenormalized band structure derived from LDA. The former leads to a small or negative plasmon dispersion, the latter to a positive dispersion (see Fig. 2 in [13]).

4. The authors claim that the plasmon is Landau damped at large q . However, looking at the $q_z=0$ data (Fig. 3), the plasmon has the same width at $q=0.66$ as it does at $q=0.08$. The authors need to explain what the basis is for the claim of Landau damping when this is clearly not reflected in the width of the peak. I recommend adding another panel to Fig. 4 showing how the width of the peak (not just its energy) depends on q , and then discussing Landau damping in terms of this width.

The change of the plasmon width is hard to see just by naked eye. In particular at $q_{||} = 0.08\text{\AA}^{-1}$ and $q_{||} = 0.66\text{\AA}^{-1}$. So we do not agree that it is not reflected in the width of the peaks. To make this more clear we have now added a new figure which shows the momentum dependence of the plasmon width data. Analogous to the plasmon energy, we have added the data from the first paper and the present paper.

5. What is the difference between χ and χ_{sc} in Eq. 3? In Nozieres and Pines, the screened χ_{sc} is the same as the Lindhard function, which is called χ_0 in this paper. Please clarify the notation.

χ_0 is the unscreened Lindhard susceptibility. χ^{sc} is the self-consistent RPA susceptibility. We have changed the notation in the revised the manuscript. $\chi^{sc} \Rightarrow \chi^{RPA}$

6. I cannot understand the meaning of this sentence: The rapid decay in the theoretical curve for the optical plasmon near $q_c = 0$ is due to the finite momentum resolution, i.e., integration of optical and acoustic plasmons within the EEL collection aperture. Why does the experimental resolution affect the theoretical curve? Please rephrase or clarify. To avoid further misunderstanding, we follow Referee 1. In the revised manuscript, we now present the curves of the Fetter model without taking into account the finite momentum resolution (solid lines). Using a dashed line we illustrate the influence of our finite momentum resolution by incorporating the integration within the EEL collection aperture in the theoretical model.

Reviewer 2

1. Although the study is well constructed and data supports the findings, I find that the impact of this paper might be lacking.

Different from Reviewer 2, we think that our paper has a strong impact on the understanding of the electronic structure of correlated electron systems. In the compound Sr_2RuO_4 we see in agreement with previous T-EELS studies on various cuprates no overdamped collective charge excitations (plasmons) for $q \lesssim q_{crit}$. Thus we come to the conclusion that spectroscopy on (high-energy) plasmons ($\omega_{pl} \gg J$) cannot support holographic theories for "strange" metals which are discussed heavily by several theoretical groups. This is at variance with R-EELS studies [3, 4, 18] where a continuum of electron-hole excitations was detected starting close to zero momentum well below the critical momentum q_c . The authors of this studies claim, that their results support the holographic theories. The difference indicates that T-EELS and R-EELS may record different response functions. This fact was discussed but could not be clarified in a recent review [19]. In the present article, we see, that the plasmon dispersion is not influenced by the normalization of the band structure at low energy detected by ARPES and optical spectroscopy. The optical plasmon dispersion can be explained by an unrenormalized band structure using an energy/temperature dependent effective mass which turns to the bare mass at high energy/temperature. Such effective mass was derived in more recent optical [5] and ARPES [7] studies and it is also supported by theoretical work [10].

Another impact is certainly the detection of acoustic plasmons in a non-cuprate system. Moreover, in our publication we show, that acoustic plasmons cannot be detected only by RIXS but also by T-EELS.

The data at 0.3 1/\AA seems to deviate strongly from the calculated curves, could the authors comment on this?

Since all data points are below the theoretical curves the deviation could result from the starting influence of the single particle continuum. However, due to the improved data analysis, the deviation from theoretical curves is lower in the revised version. All data points fit to the theoretical curves within the error bars.

The authors report the energy of the plasmons for the different q values, but it seems to me that the intensity variation might also be interesting to plot as a function of q (maybe also as a function of qa)

As a result of the improved data evaluation we added a new figure showing the intensity variation in dependence on q_{\perp} and q_{\parallel} . We furthermore calculated theoretical curves based on the dielectric function derived from Fetter [20] and explained the curves by a sum rule.

The first sentence of paragraph V seems to be lacking an end "not only"... should be followed by a but... or something similar

We thank the Reviewer. We have changed the text.

The impact of the measurements on the understanding of the correlation between acoustic, optical plasmons and superconductivity would be greatly improved by some more approachable schematic representation to guide the reader through this complex concept.

We are in some way confused. In this contribution we only deal with the strange normal state electronic structure analysed by measuring collective charge excitation using T-EELS. We have added in Fig. 1 a schematic representation of optical in-phase and acoustic out-of-phase charge oscillation in the Sr_2RuO_4 layered system.

Similarly a more visual description of the correlation between the crystal structure anisotropy (layered structure with the acoustic/optical plasmon velocity) would also help into making the work accessible to a larger community.

We have added in Fig. 1 a schematic representation of optical in-phase and acoustic out-of-phase charge oscillation in the Sr_2RuO_4 layered system.

I am missing a value of the collection angle and the acquisition time for the EELS experiments (the collection angle should give an idea of the q space resolution but this is not mentioned and might be difficult to follow for the non expert reader)

The energy and momentum resolution (we also added the corresponding collection angle) was given in Methods. We have added the acquisition times.

Reviewer 3

1. Should mass in Eq (4) be "effective mass"?

Yes, in some text books its "effective mass", in some it is "bare mass". We think it should be "effective mass" and energy dependent when calculating the low-energy plasmon energy from the band structure and the susceptibility. Well above a coupling to an excitation which dominates the correlation effects (e.g. spin excitations) one should use the unrenormalized bare mass m_b taken from a tight-binding fit of an LDA band structure [21]. This is one of the main points of the paper.

2. In Fig.4, why does the purple solid line ($q_c = 0$) dip near $q=0$? Should it not approach $\omega_P(0)$ smoothly? (See Fig 1a of Ref. 2 for example.)

See answer to Referee 1 Comments on the body of the paper 6.

3. Eq (5) deserves either more details, such as a derivation, or a reference. It is not obvious how to define "average squared Fermi velocity along the q direction"

In the revised manuscript we have given references for Eq. 5 and defined the "average squared Fermi velocity along the q direction"

4. The left side of Eq (5) gives ω_P in the limit of small q . For the range of q in the experiment, have the authors checked whether the expression is applicable? For example the authors could calculate Eq (3) and check for what range of q do the calculated peak positions agree with the expression in Eq (5).

In our first paper [13] we have calculated the dispersion using Eq. (3) and compared the experimental data. Eq. (5) agrees with the experimental data up to about $q \approx 0.3 \text{ \AA}^{-1}$. In the revised version we remark this.

Was any data collected along the 110 direction?

Yes, for the optical plasmon, we have also collected data along the 110 direction which is presented in our first paper [13]. The anisotropy is considerably smaller than in the cuprates.

References

- [1] C. G. Fatuzzo et al. "Spin-orbit-induced orbital excitations in Sr_2RuO_4 and Ca_2RuO_4 : A resonant inelastic x-ray scattering study". In: *Phys. Rev. B* 91 (15 2015), p. 155104.
- [2] Ali A. Husain et al. "Crossover of charge fluctuations across the strange metal phase diagram". In: *Phys. Rev. X* 9.4 (2019), p. 041062.
- [3] Jin Chen et al. "Consistency between reflection momentum-resolved electron energy loss spectroscopy and optical spectroscopy measurements of the long-wavelength density response of $\text{Bi}_2\text{Sr}_2\text{CaCu}_2\text{O}_{8+x}$ ". In: *Phys. Rev. B* 109 (4 2024), p. 045108.
- [4] Ali A. Husain et al. "Pines demon observed as a 3D acoustic plasmon in Sr_2RuO_4 ". In: *Nature* (2023).
- [5] D. Stricker et al. "Optical Response of Sr_2RuO_4 Reveals Universal Fermi-Liquid Scaling and Quasiparticles Beyond Landau Theory". In: *Phys. Rev. Lett.* 113.8 (2014), p. 087404.
- [6] A. Tamai et al. "High-Resolution Photoemission on Sr_2RuO_4 Reveals Correlation-Enhanced Effective Spin-Orbit Coupling and Dominantly Local Self-Energies". In: *Phys. Rev. X* 9 (2 2019), p. 021048.
- [7] A. Hunter et al. "Fate of Quasiparticles at High Temperature in the Correlated Metal Sr_2RuO_4 ". In: *Phys. Rev. Lett.* 131 (23 2023), p. 236502.

- [8] Abhishek Nag et al. “Detection of acoustic plasmons in hole-doped lanthanum and bismuth cuprate superconductors using resonant inelastic X-Ray scattering”. In: *Phys. Rev. Lett.* 125.25 (2020), p. 257002.
- [9] W. J. Padilla et al. “Constant effective mass across the phase diagram of high- T_c cuprates”. In: *Phys. Rev. B* 72 (6 2005), p. 060511.
- [10] Xiaoyu Deng et al. “How bad metals turn good: Spectroscopic signatures of resilient quasiparticles”. In: *Phys. Rev. Lett.* 110 (8 Feb. 2013), p. 086401.
- [11] C. M. Varma et al. “Phenomenology of the normal state of Cu-O high-temperature superconductors”. In: *Phys. Rev. Lett.* 63.18 (1989), pp. 1996–1999.
- [12] J. Fink. “Influence of Lifshitz transitions and correlation effects on the scattering rates of the charge carriers in iron-based superconductors”. In: *Europhys. Lett.* 113.2 (2016), p. 27002.
- [13] Martin Knupfer et al. “Propagating charge carrier plasmons in Sr_2RuO_4 ”. In: *Phys. Rev. B* 106 (24 2022), p. L241103.
- [14] G. Giuliani and G. Vignale. *Quantum Theory of the Electron Liquid*. Cambridge University Press, 2005.
- [15] H. Raether. *Excitation of plasmons and interband transitions by electrons*. Springer Verlag, Berlin, 1980.
- [16] S.E. Schnatterly. “Inelastic Electron Scattering Spectroscopy”. In: ed. by Henry Ehrenreich, Frederick Seitz, and David Turnbull. Vol. 34. Solid State Physics. Academic Press, 1979, pp. 275–358.
- [17] Jörg Fink. “Recent Developments in Energy-Loss Spectroscopy”. In: ed. by Peter W. Hawkes. Vol. 75. Advances in Electronics and Electron Physics. Academic Press, 1989, pp. 121–232.
- [18] M. Mitrano et al. “Anomalous density fluctuations in a strange metal”. In: *Proc. Natl. Acad. Sci. USA* 115.21 (2018), p. 5392.
- [19] Peter Abbamonte and Jörg Fink. “Collective charge excitations studied by electron energy-loss spectroscopy”. In: *arXiv:2404.04670* (2024).
- [20] Alexander L. Fetter. “Electrodynamics of a layered electron gas. II. Periodic array”. In: *Ann. Phys.* 88.1 (1974), pp. 1–25.
- [21] A. Liebsch and A. Lichtenstein. “Photoemission Quasiparticle Spectra of Sr_2RuO_4 ”. In: *Phys. Rev. Lett.* 84 (7 2000), pp. 1591–1594.

Author Rebuttals to the Reviewer comments

We thank the Reviewers 2 and 3 for their positive report and their recommendation for publication in Nature Communication.

Unfortunately Reviewer 1 found our reply and the increase focus on the lack of mass renormalization especially troubling. He comes to the conclusion that our results are incorrect and inconsistent not only with a large body of literature, but also with their own data. In the following we disprove this assessment point by point.

(1) Reviewer 1 claims that low-energy effects in the optical conductivity translate into high-energy effects in the loss function. We can disprove this statement by our calculations of the susceptibility and the loss function for an effective mass $m^*/m = 1$, $m^*/m = 3.5$, and an energy dependent effective mass which is below 0.2 eV 3.5 and above 0.2 eV 1. The latter energy dependent effective mass was taken from optical data on Sr_2RuO_4 [1] added by an extrapolation to higher energy. We remark that in the first round, Reviewer 3 has asked for such calculations. We referred in the the first round on our previous paper [2] where we have shown data for a constant effective mass. In these calculations we receive a positive plasmon dispersion for an unrenormalized band structure. Using a constant effective mass $m^*/m_0 = 3.5$, the calculations yield a considerably smaller dispersion. This is in agreement with the prediction, that a constant large effective mass leads to a negative dispersion [3]. This prediction was experimentally detected in Cs metal, where an admixture of d bands leads to a large effective mass [4]. See also theoretical work on this topic [5, 6]. Recently we have also calculated the susceptibility χ_0 and the loss function for an energy dependent effective mass taken from optical spectroscopy (see above). In the susceptibility, we see a flattening of the dispersion in the low (ω, q_{\parallel}) range. Although there is a strong influence of the low energy mass enhancement ($m^*/m_0 = 3.5$) on the susceptibility, there is a negligible transfer to the high energy plasmon dispersion because in the high energy (ω, q_{\parallel}) range, the mass renormalization is zero ($m^*/m_0 = 1$). We therefore stress, that although the susceptibility and thus the optical conductivity in the low (ω, q) range is changed when there is mass enhancement at low energies, the optical plasmon dispersion is almost not influenced. As the results on Sr_2RuO_4 were conflated with those on cuprates by other groups, we dare to explain our results on the unrenormalized plasmon dispersion in cuprates [3, 7, 8, 9] in a similar scenario.

(2) Because of the observed finite width at zero momentum, Reviewer 1 is deeming that correlation effects must be important in our measurements of the plasmon dispersion. This problem has been addressed in a very early paper in Ref. [10]. There, the authors concluded that the finite plasmon width at small q is not caused by a decay into electron hole excitations. It is thus not related to finite lifetime of quasi particles, i.e., to the imaginary part of the self energy. Later, the finite width at small q was theoretically explained by interband transition due to a back-folding of the conduction band from neighboring Brillouin zones (BZ) [11]. This explanation was experimentally confirmed

by EELS studies on the plasmon width in alkali metals [4, 12]: there it was shown that the plasmon width is proportional to the squared pseudopotential which is reflecting the unoccupied part of the conduction band into the first BZ. Thus, the plasmon width is caused by a decay of the plasmon into inter-band transitions. See also the recent study of the line width of acoustic plasmons in p- and n-type doped cuprates [13]

(3) Reviewer 1 complains that the RPA theory grossly deviates of the plasmon dispersion from the experimental data in Fig. 4 for $q_{\parallel} > 0.3 \text{ \AA}^{-1}$. We remark that the calculated plasmon dispersion is an expansion of the Fetter model for low momentum to high q_{\parallel} regions. We have noted this in the revised manuscript. The lowering of the plasmon energy above q_{crit} was already discussed in the first version of the manuscript and in our previous paper [2]. There we also mentioned that this lowering has been detected also in Al [14]. Finally, we mention that the lowering of the maximum in the loss function is also seen in our calculations presented in the revised manuscript in Fig. 5: the maximum of the collective excitations (plasmons) transforms at q_{crit} into the maximum of the decaying plasmon and the particle-hole excitations which appears at lower energies.

(4) Reviewer 1 notes that the different calculations of charge spectrum between our work and that in Nature 621, 66 (2023) cannot traced back to different effective masses used in the calculations. We point out that our calculations of the loss function, based on an unrenormalized band structure yields a positive dispersion while a renormalized band structure yields a considerably smaller dispersion. Possibly EELS in reflection measures a different response function when compared with EELS in transmission.

(4) Reviewer 1 mentions a conflict between a recent RIXS paper in which a mass enhancement of about 2 is discussed for the acoustic plasmon. The citations originate from parts of this paper and are taken out of the context. They are taken from paragraphs which deal with acoustic plasmons. Other parts of that paper clearly point out that using an energy dependent effective mass, an unrenormalized optical plasmon dispersion can be rationalized.

References

- [1] D. Stricker et al. “Optical Response of Sr_2RuO_4 Reveals Universal Fermi-Liquid Scaling and Quasiparticles Beyond Landau Theory”. In: *Phys. Rev. Lett.* 113.8 (2014), p. 087404. URL: <https://link.aps.org/doi/10.1103/PhysRevLett.113.087404>.
- [2] Martin Knupfer et al. “Propagating charge carrier plasmons in Sr_2RuO_4 ”. In: *Phys. Rev. B* 106 (24 2022), p. L241103. DOI: 10.1103/PhysRevB.106.L241103. URL: <https://link.aps.org/doi/10.1103/PhysRevB.106.L241103>.
- [3] V. G. Grigoryan, G. Paasch, and S.-L. Drechsler. “Determination of an effective one-electron spectrum from the plasmon dispersion of nearly optimally doped

- Bi₂Sr₂CaCu₂O₈”. In: *Phys. Rev. B* 60.2 (1999), pp. 1340–1348. URL: <https://link.aps.org/doi/10.1103/PhysRevB.60.1340>.
- [4] A. vom Felde, J. Sprösser-Prou, and J. Fink. “Valence-electron excitations in the alkali metals”. In: *Phys. Rev. B* 40.15 (1989), pp. 10181–10193. URL: <https://link.aps.org/doi/10.1103/PhysRevB.40.10181>.
- [5] F. Aryasetiawan and K. Karlsson. “Energy Loss Spectra and Plasmon Dispersions in Alkali Metals: Negative Plasmon Dispersion in Cs”. In: *Phys. Rev. Lett.* 73 (12 Sept. 1994), pp. 1679–1682. DOI: 10.1103/PhysRevLett.73.1679. URL: <https://link.aps.org/doi/10.1103/PhysRevLett.73.1679>.
- [6] G. Paasch and V.G. Grigoryan. “Quasi-classical theory of the alkali metal plasmon dispersion”. In: *Ukr. J. Phys.* 44 (1999), p. 1480.
- [7] N. Nücker et al. “Plasmons and interband transitions in Bi₂Sr₂CaCu₂O₈”. In: *Phys. Rev. B* 39.16 (1989), pp. 12379–12382. URL: <https://link.aps.org/doi/10.1103/PhysRevB.39.12379>.
- [8] N. Nücker et al. “Long-wavelength collective excitations of charge carriers in high- T_c superconductors”. In: *Phys. Rev. B* 44.13 (1991), pp. 7155–7158. DOI: 10.1103/PhysRevB.44.7155.
- [9] Friedrich Roth et al. “Evidence for an orbital dependent Mott transition in the ladders of (La, Ca)_xSr_{14-x}Cu₂₄O₄₁ derived by electron energy loss spectroscopy”. In: *Phys. Rev. B* 101.19 (2020), p. 195132. URL: <https://link.aps.org/doi/10.1103/PhysRevB.101.195132>.
- [10] D. F. DuBois and M. G. Kivelson. “Electron Correlational Effects on Plasmon Damping and Ultraviolet Absorption in Metals”. In: *Physical Review* 186.2 (1969), pp. 409–419. DOI: 10.1103/physrev.186.409.
- [11] G. Paasch. “Influence of Interband Transitions on Plasmons in the Alkali Metals: Pseudopotential Calculation”. In: *phys. stat. sol.* 38 (1970), K123.
- [12] P. C. Gibbons and S. E. Schnatterly. “Comment on the line shape of the plasma resonance in simple metals”. In: *Phys. Rev. B* 15.4 (1977), pp. 2420–2421. URL: <https://link.aps.org/doi/10.1103/PhysRevB.15.2420>.
- [13] Abhishek Nag et al. *Impact of electron correlations on two-particle charge response in electron- and hole-doped cuprates*. 2024. arXiv: 2407.15692 [cond-mat.str-el]. URL: <https://arxiv.org/abs/2407.15692>.
- [14] PE Batson and J Silcox. “Experimental energy-loss function, $\text{Im}[-1/\varepsilon(q, \omega)]$, for aluminum”. In: *Phys. Rev. B* 27.9 (1983), p. 5224.

Dear Editor,

Thank for your reply and the referee reports. We thank reviewer 4 for the positive report and its comment on the potentially different behavior of the surface of Sr_2RuO_4 . We comment on this and on the points raised by referee 1 point by point below.

Referee 1:

Regarding their rebuttal, the authors argue in point (1) against my statement that the mass renormalization found at low energy will affect the plasmon dispersion. They do so by producing a new figure 5 including RPA calculations for three model effective masses of the carriers. They find that introducing a smooth dependence from a renormalized mass of 3.5 to an unrenormalized mass at high energy yields negligible changes of the plasmon dispersion compared with a fully unrenormalized mass. This argument is clever but lacks a physical explanation to ground what otherwise remains an ad hoc feature to reproduce the data. In this sense the authors logic is circular. In addition, Fig. 5 makes clear that their plasmon dispersion from Fig. 4 is much more similar to the one they obtain for $m^/m=3.5$ (minus the qz dependence from the layered nature of the electron gas). I would encourage the author to plot the two dispersions on top of the data and see which fits best. It is unclear to this reviewer how the authors can reconcile the dispersion found in Fig 4 with their core claim. However, the magnitude of this discrepancy in the economy of the claim and with rest of the optics/ARPES/RIXS literature is too big to ignore.*

Answer:

Referee 1 states that the use of an energy dependent effective mass is clever but lacks a physical explanation and that it remains an ad hoc feature to reproduce the data. We do not understand this statement at all. Already in the middle of last century Engelsberg and Schrieffer [1] calculated the self energy for a similar case, the electron phonon interaction. Following this work, in a standard solid state textbook it was concluded that well above the Debye energy the phonon correction of the band structure becomes insignificant. In correlated metals similar results were reported, probably not caused by electron-phonon coupling, but possibly by a coupling to spin fluctuations. We again stress that we use experimental optical data from the Dirk van der Marel group [3] to model the energy dependent effective mass. We also stress that the energy dependent mass in correlated systems and the detection of resilient quasiparticles at high energies (e.g. the optical plasmon energy) was predicted by DFT+DMFT calculations [4]. Finally, we mention that already the marginal Fermi liquid theory [5, 6] is based on a diverging effective mass at low energy which transforms at higher energies into an unrenormalized effective mass. Thus we strongly contradict that our logic is circular and an ad hoc feature.

In the revised version, we followed the advice of Referee 1 and compared in Fig. 5, fourth column the experimental optical plasmon dispersion with the calculated disper-

sion. There is a considerably better agreement of the experimental data with the theoretical calculations for $m^*/m_0 = 1$ (black solid line) compared to those for $m^*/m_0 = 3.5$.

Regarding point 2, the reference to interband transitions is interesting, but needs to be substantiated. The authors write in the paper that Already in our previous paper [36], we have emphasized that the width is almost constant in the studied low momentum range. This means that it is not related to electron-electron interaction which would lead to a quadratic increase of the width as a function of momentum transfer [50]. The momentum-dependent width in Fig. 6 looks quadratic to me, and I encourage the authors to deepen the discussion of this issue if they want to truly settle the issue correlations vs interband transitions. At this stage, the paper says too little to address the matter and the arguments in favor of interband transitions must be taken at authors word.

Answer:

Fig. 6 of the previous version shows a clear plateau ,i.e., constant width of the plasmon peaks for $q_{\parallel} < q_{\text{crit}} \approx 0.4 \text{ \AA}^{-1}$. q_{crit} agrees roughly with the momentum at which the optical plasmon merges into the calculated Landau continuum (see Fig. 4). Furthermore, if the band structure would be renormalized at high energy, the energy of the continuum would be reduced. In this case q_{crit} would be enhanced or the plasmon dispersion would no more merge into the continuum (see Fig. 5, row two, column 3 and 4).

Point 3 does not yet address the fundamental discrepancy between experiment and unrenormalized mass RPA, in the sense that the RPA provided by the authors in Fig. 5 convinced me even more that the mass is renormalized, as they see a dispersion in Fig. 4 akin to the unlabeled panel in second row, third column of Fig. 5.

Answer:

Fig. 5, fourth column reveals larger discrepancy of the experimental data to the calculated renormalized $m^*/m_0 = 3.5$ case compared to data using an unrenormalized band structure.

Regarding point 4 of the rebuttal, I just note that, in addition to the irreducible incompatibility with the RPA in Husain et al, the RPA of this manuscript is also incompatible with calculations recently done by the Cohen group PRB 110, 155127 (2024).

Answer:

The paper by the Cohen group [7] deals with the question, whether the demon mode in Sr_2RuO_4 can contribute to the pairing interaction of electrons. As shown in the R-EELS work of Husain [8] and in the calculations of the Cohen group [7] the demon plasmon occurs below 0.2 eV. According to the optical data [3] the average mass is strongly enhanced in the low energy region while according to our calculations the high energy

optical plasmon dispersion is related to an unrenormalized effective mass. This may be one reason of the incompatibility of our calculations with those of the Cohen group. Another reason may be that in the calculations of the Cohen group, the Coulomb potential of the layered structure in Sr_2RuO_4 has not been taken into account in the calculations of the dielectric function.

Finally, concerning point 5, I will leave to the editor and the readers to judge if it is a sustainable position to advocate for a lack of mass renormalization here while invoking mass renormalization to describe plasmon dispersion and width in <https://arxiv.org/pdf/2407.15692> at the same time. The authors suggest I have taken sentences out of context from their other work. However, I will note that even the abstract of that work states The t-J -V model, which includes the correlation effects implicitly, accurately describes the plasmon dispersions as resonant excitations outside the single-particle intra-band continuum. In comparison, a quantitative description of the plasmon dispersion in the RPA approach is obtained only upon explicit consideration of re-normalized electronic band parameters. Our comparative analysis shows that electron correlations significantly impact the low-energy plasmon excitations across the cuprate doping phase diagram, even at long wavelengths. Thus, complementary information on the evolution of electron correlations, influenced by the rich electronic phases in condensed matter systems, can be extracted through the study of two-particle charge response. The claim contained even in the abstract of that work is in conflict with the message of the paper under consideration here. The authors try to explain this inconsistency away by talking about acoustic plasmons, but, by using their own words, the Sr_2RuO_4 plasmons at selected qz are also acoustic, hence squarely within the scope of the other work. I maintain that the key thesis of the present manuscript is questionable and at odds with decades of ARPES, optics, RIXS, as well as in clear contrast with the authors data.

Answer:

The RIXS paper [9] discusses acoustic plasmons in Sr_2RuO_4 . These occur at low energy. Optical plasmons could not be reached. Therefore, the plasmon velocity is determined by the renormalized band structure. Because it is the main point of that paper, this is mentioned in the abstract. On the other hand, in the discussion of that paper, our T-EELS paper is mentioned to emphasize the difference between low energy and high energy plasmon dispersion in correlated metals. This is different to nearly-free electron metals, where the renormalization range of the bands (by plasmon) is comparable to the plasmon energy. In the present paper, our energy resolution was not high enough to obtain reasonably data of the plasmon velocity. This is mentioned in the manuscript and we refer to future EELS data derived with improved energy resolution.

Referee 4:

One point that it is worth making, given the wide interest of the material, is to connect and comment on the results with the view that the surface of Sr_2RuO_4 behaves

differently, due to energetically favorable rotation of octahedra (see for example Nature Communications 15 (1), 9521 (2024) and references therein). A simple argument based on the nature of the experiments will be enough to make the distinction, and I believe it will be informative to the readers.

Answer:

We thank Referee 4 to draw our attention to that paper.

Sincerely yours

Johannes Schultz on behalf of the authors

Performed changes (marked in red)

We emphasize that the performed minor changes do not modify our essential results. Rather these changes which are partially induced by the comments of the referees support our argumentation.

page 2:

We have introduced two references which deal with the problems of surfaces preparation in Sr_2RuO_4 .

page 5 and 7 left column:

We have rewritten the discussion on the optical plasmon dispersion and the comparison with calculations. First we deal with calculations using the Fetter-Apostol model. This part is essentially equal to the discussion in the previous version of our manuscript. The second part of the discussion of the plasmon dispersion is based on our calculations of the momentum dependent loss function. In the revised version, we discuss in more detail the non-sensitivity of the high-energy optical plasmon dispersion on the low-energy band renormalization.

Fig. 5:

In the revised version in column five we now compare the experimental optical plasmon dispersion with the calculated one for $m^*/m_0 = 1$ and $m^*/m_0 = 3.5$.

page 7 right column:

We have slightly changed the discussion of the plasmon width. In Fig. 6 we now compare the experimental data with the width derived from the calculated momentum dependent loss function.

References

- [1] S. Engelsberg and J. R. Schrieffer. “Coupled Electron-Phonon System”. In: *Phys. Rev.* 131 (3 Aug. 1963), pp. 993–1008. DOI: 10.1103/PhysRev.131.993. URL: <https://link.aps.org/doi/10.1103/PhysRev.131.993>.
- [2] N. W. Ashcroft and N. D. Mermin. *Solid State Physics*. Saunders College, 1976.
- [3] D. Stricker et al. “Optical Response of Sr_2RuO_4 Reveals Universal Fermi-Liquid Scaling and Quasiparticles Beyond Landau Theory”. In: *Phys. Rev. Lett.* 113.8 (2014), p. 087404. URL: <https://link.aps.org/doi/10.1103/PhysRevLett.113.087404>.
- [4] Xiaoyu Deng et al. “How bad metals turn good: Spectroscopic signatures of resilient quasiparticles”. In: *Phys. Rev. Lett.* 110 (8 Feb. 2013), p. 086401. DOI: 10.1103/PhysRevLett.110.086401. URL: <https://link.aps.org/doi/10.1103/PhysRevLett.110.086401>.
- [5] C. M. Varma et al. “Phenomenology of the normal state of Cu-O high-temperature superconductors”. In: *Phys. Rev. Lett.* 63.18 (1989), pp. 1996–1999. URL: <https://link.aps.org/doi/10.1103/PhysRevLett.63.1996>.
- [6] J. Fink. “Influence of Lifshitz transitions and correlation effects on the scattering rates of the charge carriers in iron-based superconductors”. In: *Europhysics Letters* 113.2 (2016), p. 27002. DOI: 10.1209/0295-5075/113/27002. URL: <https://dx.doi.org/10.1209/0295-5075/113/27002>.
- [7] Young Woo Choi, Jisoon Ihm, and Marvin L. Cohen. “Pairing interaction from three-dimensional acoustic plasmon demon modes in Sr_2RuO_4 ”. In: *Phys. Rev. B* 110 (15 Oct. 2024), p. 155127. DOI: 10.1103/PhysRevB.110.155127. URL: <https://link.aps.org/doi/10.1103/PhysRevB.110.155127>.
- [8] Ali A. Husain et al. “Pines demon observed as a 3D acoustic plasmon in Sr_2RuO_4 ”. In: *Nature* (2023). ISSN: 1476-4687. DOI: 10.1038/s41586-023-06318-8. URL: <https://doi.org/10.1038/s41586-023-06318-8>.
- [9] Abhishek Nag et al. “Impact of electron correlations on two-particle charge response in electron- and hole-doped cuprates”. In: *Phys. Rev. Res.* 6 (4 Nov. 2024), p. 043184. DOI: 10.1103/PhysRevResearch.6.043184. URL: <https://link.aps.org/doi/10.1103/PhysRevResearch.6.043184>.